# Sleep-dependent upscaled excitability, saturated neuroplasticity, and modulated cognition in the human brain

**Mohammad Ali Salehinejad[1,2], Elham Ghanavati[1,3], Jörg Reinders[4], Jan G Hengstler[4], Min-Fang Kuo[1], Michael A Nitsche[1,5]\***

[1]Department of Psychology and Neurosciences, Leibniz Research Centre for Working Environment and Human Factors, Dortmund, Germany; [2]International Graduate School of Neuroscience, Ruhr-University Bochum, Bochum, Germany; [3]Department of Neuropsychology, Institute of Cognitive Neuroscience, Faculty of Psychology, Ruhr University Bochum, Bochum, Germany; [4]Department of Toxicology, Leibniz Research Centre for Working Environment and Human Factors, Dortmund, Germany; [5]Department of Neurology, University Medical Hospital Bergmannsheil, Bochum, Germany

**\*For correspondence:** nitsche@ifado.de

**Abstract** Sleep strongly affects synaptic strength, making it critical for cognition, especially learning and memory formation. Whether and *how* sleep deprivation modulates human brain physiology and cognition is not well understood. Here we examined how overnight sleep deprivation vs overnight sufficient sleep affects (a) cortical excitability, measured by transcranial magnetic stimulation, (b) inducibility of long-term potentiation (LTP)- and long-term depression (LTD)-like plasticity via transcranial direct current stimulation (tDCS), and (c) learning, memory, and attention. The results suggest that sleep deprivation upscales cortical excitability due to enhanced glutamate-related cortical facilitation and decreases and/or reverses GABAergic cortical inhibition. Furthermore, tDCS-induced LTP-like plasticity (anodal) abolishes while the inhibitory LTD-like plasticity (cathodal) converts to excitatory LTP-like plasticity under sleep deprivation. This is associated with increased EEG theta oscillations due to sleep pressure. Finally, we show that learning and memory formation, behavioral counterparts of plasticity, and working memory and attention, which rely on cortical excitability, are impaired during sleep deprivation. Our data indicate that upscaled brain excitability and altered plasticity, due to sleep deprivation, are associated with impaired cognitive performance. Besides showing how brain physiology and cognition undergo changes (from neurophysiology to higher-order cognition) under sleep pressure, the findings have implications for *variability* and optimal *application* of noninvasive brain stimulation.

## Editor's evaluation

This paper provides a comprehensive investigation into the neural effects of sleep deprivation in humans across a broad range of methods and, using non–invasive brain stimulation as well as electrophysiological markers and behavioral measures, the study demonstrates that sleep deprivation results in higher cortical excitability, which may explain the negative impact of sleep deprivation on cognitive processes.

## Introduction

Over the past decade, a strong link has been established between sleep and cognition (*Yaffe et al., 2014*; *Lowe et al., 2017*). Adequate sleep is critical for optimal cognitive functions across the lifespan (*Carskadon, 2011*; *Lo et al., 2016*). Findings from experimental settings support this critical role of sleep for cognition in animals (*Rasch and Born, 2013*; *Boyce et al., 2017*) and humans (*Krause et al., 2017*) especially for memory consolidation and sequence learning (*Stickgold, 2005*; *Chouhan et al., 2021*). As a ubiquitous *physiological* phenomenon, sleep has extensive impacts on brain physiology and especially on parameters relevant to cognition such as brain excitability and plasticity.

Previous experimental studies, mostly in nonhuman animals, linked sleep with synaptic homeostasis. Specifically, extended wakefulness (or sleep deprivation) is associated with the expression of long-term potentiation (LTP)-related molecular changes and plasticity-related genes (e.g. BDNF, CREB) in the brain, leading to saturation of synaptic potentiation in both *Drosophila* and mice models (*Tononi and Cirelli, 2003*; *Bushey et al., 2011*). Sleep, on the other hand, desaturates synapses that have been potentiated during wakefulness in mice (*Vyazovskiy et al., 2008*; *Miyamoto et al., 2021*) resulting in a renewed capacity for encoding new information. Recent studies confirmed this sleep-dependent synaptic downscaling by showing reduced or weakened synaptic connections in the primary motor and somatosensory cortex of mice during sleep (*de Vivo et al., 2017*; *Miyamoto et al., 2021*). This demonstrates that sleep is required for preparing the brain for proper cognitive, motor, and physiological functioning; however, the effect of sleep on specific parameters of *human* brain physiology and their association with cognition and behavior remains to be further determined.

In humans, molecular mechanisms of synaptic homeostasis cannot be directly studied; however, non-invasive (indirect) markers of brain physiology can be used for studying the impact of sleep and extended wakefulness on synaptic potentiation and cortical excitability. Non-invasive brain stimulation (NIBS) techniques are safe methods for monitoring and modifying brain functions in humans providing a means for studying the causality of brain-behavior relationships (*Polanía et al., 2018*). Several NIBS techniques, including transcranial magnetic stimulation (TMS) and transcranial electrical stimulation (tES), are widely used to non-invasively monitor and induce changes in cortical excitability and neuroplasticity (*Nitsche and Paulus, 2000*; *Huang et al., 2017*; *Polanía et al., 2018*). It is shown that corticospinal excitability increases after sleep deprivation (*Kuhn et al., 2016*; *Ly et al., 2016*). This increase of brain excitability comes with a reduced inhibitory control mechanism in humans as well (*Kreuzer et al., 2011*; *Placidi et al., 2013*) which can reduce the ability of the brain to induce neuroplasticity (as a result of synaptic saturation). In this line, decreased LTP has been shown in rats after sleep deprivation in both, in vivo and in vitro (*Kopp et al., 2006*; *Vyazovskiy et al., 2008*; *Zhou et al., 2020*), and a recent human study also showed decreased LTP-like plasticity, induced by paired associative stimulation (*Kuhn et al., 2016*).

The number of available studies about the impact of sleep deprivation on human brain physiology relevant for cognitive processes is limited, and knowledge is incomplete. With respect to cortical excitability, *Kuhn et al., 2016* showed increased excitability under sleep deprivation via a global measure of corticospinal excitability, the TMS intensity needed to induce motor-evoked potentials (MEPs) of a specific amplitude. Specific information about the cortical systems, including neurotransmitters and neuromodulators involved in these effects (e.g. glutamatergic, GABAergic, and cholinergic) is, however, missing. The level of cortical excitability affects neuroplasticity too, a relevant physiological derivate of learning and memory formation. While sleep deprivation-dependent alteration of LTP-like plasticity in humans was recently investigated (*Kuhn et al., 2016*), the effects of sleep deprivation on long-term depression (LTD)-like plasticity, which is required for a complete picture are, however, not explored so far. In the present study, we aimed to complete the current knowledge and also monitor basic cognitive abilities that critically depend on cortical excitability (working memory and attention) and neuroplasticity (motor learning) to gain mechanistic knowledge about sleep deprivation-dependent performance decline. Finally, we aimed to explore if the impact of sleep deprivation on brain physiology and cognition differs from the effects of non-optimal time of day performance in different chronotypes, which we recently explored in a parallel study with an identical experimental design (*Salehinejad et al., 2021*). The use of measures of different modalities in this study allows us to comprehensively investigate the impact of sleep deprivation on brain and cognitive functions which is largely missing in the human literature.

In addition to these primary objectives, we were interested in brain oscillatory activities which are well-established indicators of the sleep-wake cycle and can inform us about the physiological state of the sleep-deprived brain. Specifically, theta oscillations are related to sleep and cognition. Here, at least two types of theta oscillations are distinguishable: one related to cognition and information processing, which occurs during wakefulness, but also rapid eye movement (REM) sleep (*Brown et al., 2012*; *Puentes-Mestril et al., 2019*), and one related to sleep pressure due to extended wakefulness (*Vyazovskiy and Tobler, 2005*). For the former, animal studies show that it is generated by the hippocampus and involves mainly the temporal lobes at the level of the neocortex. In humans, where the temporal lobes are located ventrally and thus difficult to record specifically from surface EEG, these theta rhythms with a strong regularity are observed mainly in frontal and midline cortices. The second type of theta, which is of main interest here, is of cortical origin, less regular, predominantly but not exclusively observed over frontal-midline areas and builds up with growing sleep pressure, including sleep deprivation, in both, animals and humans studies (*Finelli et al., 2000*; *Vyazovskiy and Tobler, 2005*; *Brown et al., 2012*; *Magnuson et al., 2022*; *Snipes et al., 2022*). While there is no clear and direct link between this sleep pressure-dependent theta activity and synaptic strength as suggested by some works, recent works in humans, however, linked it to an increase of cortical excitability (*Kuhn et al., 2016*).

In the sleep deprivation paradigm applied in the present study, participants are kept in an extended wakefulness condition for a certain amount of time. We accordingly investigated the impact of one-night sleep deprivation, compared to one-night sufficient sleep on non-invasive parameters of human brain physiology and cognitive performance. Specifically, we monitored cortical excitability of the brain via TMS protocols that measure cortical inhibition and facilitation (i) and induced both LTP-like and LTD-like plasticity (ii). We expected increased cortical excitability (specifically enhanced glutamate-related intracortical facilitation (ICF) and decreased GABA-related intracortical inhibition), and a resultant saturated state for plasticity induction, leading to lower plasticity induction for both, LTP- and LTD-like plasticity under sleep deprivation. We furthermore examined learning and memory formation as behavioral indices of brain plasticity (iii), and working memory and attention which depend on cortical excitability (iv) along with their electrophysiological correlates. Here we expected compromised performance and lower amplitude event-related potential (ERP) components under sleep deprivation. We also assessed resting-EEG theta/alpha, as an indirect measure of homeostatic sleep pressure (*Vyazovskiy and Tobler, 2005*; *Leemburg et al., 2010*) and examined cortisol and melatonin concentration to see how these are affected under sleep conditions, given the reported mixed effects in previous studies.

To do so, we recruited 30 healthy, right-handed participants in this randomized, crossover study. All participants attended two experimental sessions after having sufficient sleep (23:00–8:00), or sleep deprivation (23:00–8:00). All physiological, behavioral, and hormonal measures were obtained in each session (see Methods for details) at a fixed time. For the neuroplasticity measures, half of the participants received anodal and the other half cathodal stimulation in a randomized, sham-controlled parallel-group design. *Figure 1* shows the detailed course of study.

## Results

### Sleep deprivation upscales cortical excitability

We monitored corticospinal and intracortical excitability of the motor cortex after 'sufficient sleep' and 'sleep deprivation' sessions with different TMS protocols. Input-output curve (I-O curve) and ICF were used as measures of global corticospinal excitability and cortical facilitation, respectively. Short-interval cortical inhibition (SICI), I-wave facilitation, and short-latency afferent inhibition (SAI) were applied as cortical inhibition protocols. These TMS protocols are based on different predominant neurotransmitter systems related to cortical facilitation (glutamatergic) and inhibition (GABAergic, cholinergic; *Chen, 2000*; *Di Lazzaro et al., 2000*; *Di Lazzaro et al., 2005a*; see Methods). Baseline MEP values of control conditions in TMS protocols did not significantly differ across sleep conditions (*Supplementary file 1A and B*), and the changes in protocol-specific MEPs cannot be due to baseline MEP differences across sleep conditions.

#### Input-output curve

I-O curve is a global measure of *corticospinal* excitability (*Boroojerdi et al., 2001*) and the slope of the I-O curve reflects excitability of corticospinal neurons modulated by glutamatergic activity at higher

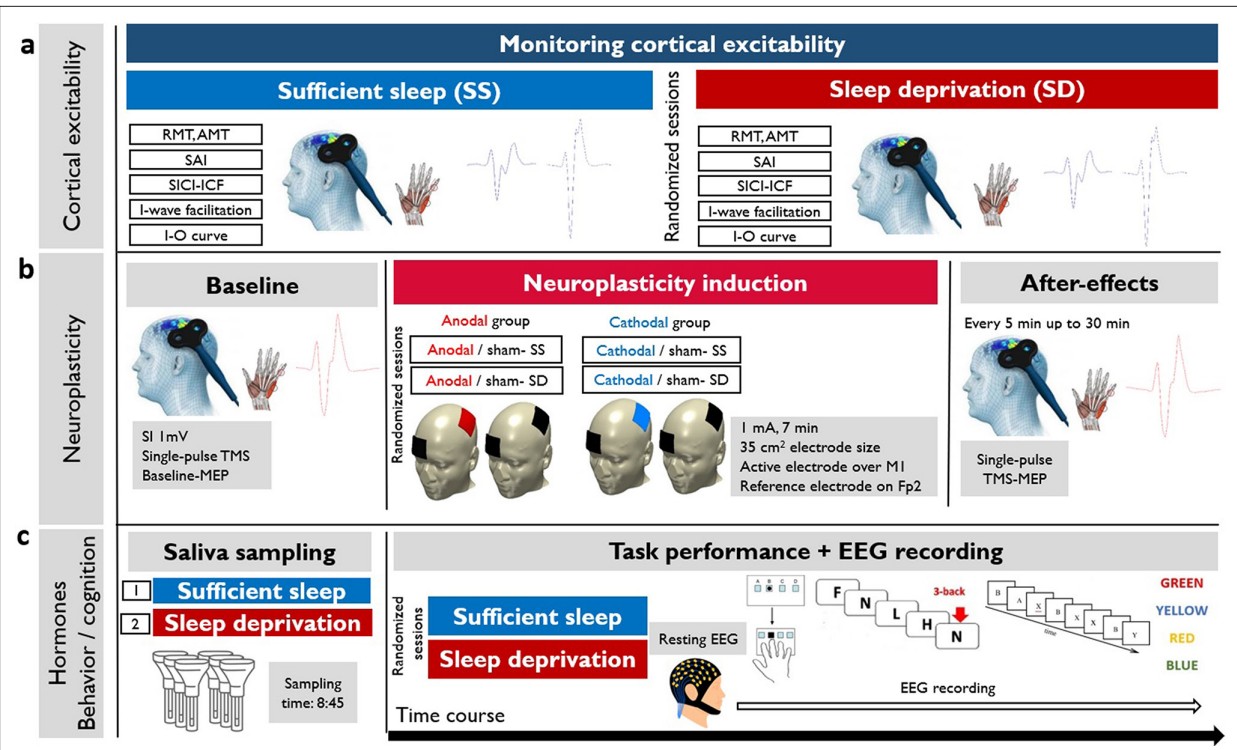

**Figure 1.** The course of the experiment. We recruited 30 young healthy participants to attend two experimental sessions in a randomized order (sufficient sleep and overnight sleep deprivation). (**a**) Using single-pulse and double-pulse transcranial magnetic stimulation (TMS) protocols, corticospinal and corticocortical excitability were measured after sleep deprivation or sufficient sleep. RMT: resting motor threshold; AMT: active motor threshold; SAI: short-latency afferent inhibition; SICI-ICF: short-latency intracortical inhibition and facilitation; I-O curve: input-output curve. (**b**) Neuroplasticity was induced with anodal and cathodal stimulation in two parallel groups (anodal vs cathodal group) after sufficient sleep (SS) vs sleep deprivation (SD). SI1 mv: stimulation intensity to elicit a motor evoked potential (MEP) amplitude of 1 mV, M1: primary motor cortex; Fp2: right supraorbital area. (**c**) Saliva samples were taken at 8:45 in each session. Following the resting-EEG acquisition, participants performed motor learning, working memory, and attention tasks at the beginning of each experimental session (sufficient sleep vs sleep deprivation) while their EEG was recorded. SRTT: serial reaction time task, AX-CPT: AX continuous performance task.

TMS intensities (see Methods). The results of the 2×4 ANOVA showed a marginally significant interaction of sleep condition × TMS intensity ($F_{1.71}$=3.41, p=0.048; $\eta p^2$=0.10), and significant main effects of sleep condition ($F_1$=4.95, p=0.034; $\eta p^2$=0.14) and TMS intensity ($F_{1.22}$=100.13, p<0.001; $\eta p^2$=0.77) on the slope of the I-O curve. MEP amplitudes were numerically larger at all TMS intensities after sleep deprivation vs sufficient sleep; however, these differences were not significantly based on the Bonferroni-corrected post hoc comparisons (*Figure 2a*).

### Short-latency intracortical inhibition and intracortical facilitation

In this double-pulse TMS protocol, the inter-stimulus interval (ISI) between a subthreshold conditioning stimulus and a suprathreshold test stimulus determines inhibitory (ISIs 2 and 3 ms) or facilitatory (ISIs 10 and 15 ms) effects on cortical excitability (*Kujirai et al., 1993*). The results of the 2×5 ANOVA showed a significant interaction of sleep condition × ISI ($F_{3.69}$=14.85, p<0.001, $\eta p^2$=0.34), and significant main effects of sleep condition ($F_1$=13.81, p<0.001, $\eta p^2$=0.72), and ISI ($F_{3.10}$=93.77, p<0.001, $\eta p^2$=0.76) on MEP values. Bonferroni-corrected post hoc comparisons revealed a significant intracortical inhibition shown by decreased MEPs in the ISI 2 and 3 ms conditions only after sufficient sleep, and significant differences of MEPs obtained with these ISIs across sleep conditions (*Figure 2b*). This indicates that intracortical inhibition was significantly lower after sleep deprivation vs sufficient sleep. For ICF, MEP amplitudes were significantly enhanced only at an ISI of 15 ms when compared with single pulse-elicited MEP amplitudes (baseline) after sufficient sleep, while they were significantly increased at ISIs of 10 and 15 ms after sleep deprivation. These MEPs were also significantly larger

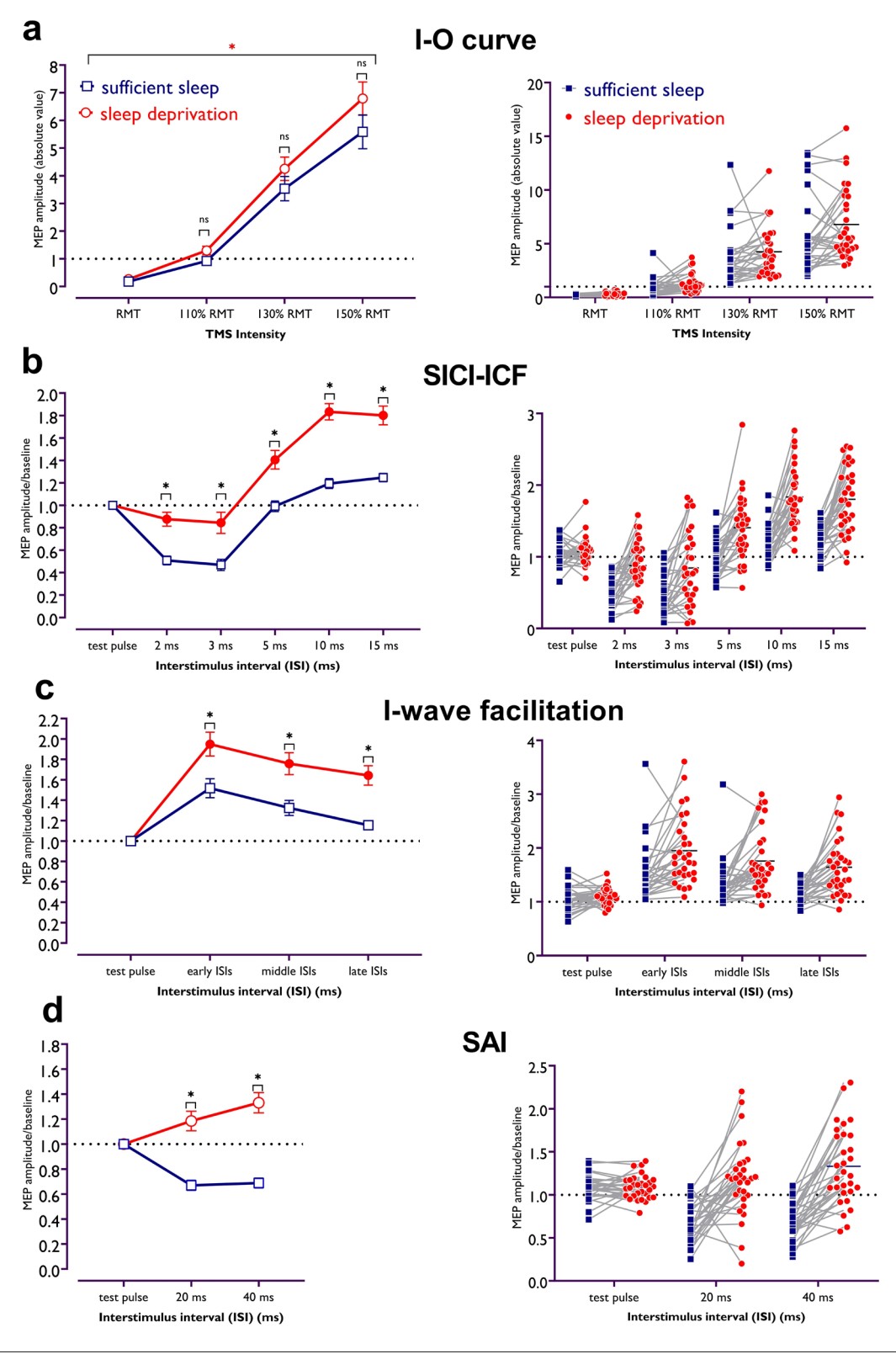

**Figure 2.** Corticospinal and corticocortical excitability after sufficient sleep vs sleep deprivation. (a) There is a trend of higher corticospinal excitability after the sleep deprivation session compared to sufficient sleep, especially at 150% of resting motor threshold (RMT) intensity. The red asterisk refers to significant effects of sleep condition (p=0.034) and transcranial magnetic stimulation (TMS) intensity (p<0.001). (b) Cortical inhibition

*Figure 2 continued on next page*

*Figure 2 continued*

significantly decreased after sleep deprivation as compared with sufficient sleep ($t_{ISI2}$=4.24, p<0.001; $t_{ISI3}$=4.50, p<0.001). In contrast, cortical facilitation is significantly upscaled after sleep deprivation compared with sufficient sleep ($t_{ISI10}$=7.69, p<0.001; $t_{ISI15}$=6.66, p<0.001). (**c**) I-wave peaks were significantly facilitated for early and middle inter-stimulus intervals (ISIs) after both, sufficient sleep and sleep deprivation, and for late ISIs only after sleep deprivation. For all ISIs, I-wave peaks were significantly more upscaled after sleep deprivation vs sufficient sleep ($t_{early}$=3.90, p<0.001; $t_{middle}$=3.91, p<0.001; $t_{late}$=4.40, p<0.001), indicative of less cortical inhibition. (**d**) Cortical inhibitory effect of peripheral nerve stimulation on motor cortical excitability was observed only after sufficient sleep ($t_{ISI20}$=4.53, p<0.001; $t_{ISI40}$=4.25, p<0.001), whereas the inhibitory effect of peripheral stimulation turned to excitatory effects after sleep deprivation ($t_{ISI20}$=2.54, p=0.035; $t_{ISI40}$=4.55, p<0.001). Motor-evoked potential (MEP) amplitude was significantly upscaled after sleep deprivation vs sufficient sleep ($t_{ISI20}$=7.08, p<0.001; $t_{ISI40}$=8.83, p<0.001). All pairwise comparisons were calculated using the Bonferroni correction for multiple comparisons (n=30). All error bars represent the s.e.m. Filled symbols represent a significant difference in MEP amplitudes compared to the respective test pulses (for short-latency intracortical inhibition and intracortical facilitation [SICI-ICF], I-wave, short-latency afferent inhibition [SAI]) or MEP at RMT intensity (for input-output curve [I-O curve]). Asterisks represent statistically significant comparisons between sleep conditions. ms: milliseconds.

after sleep deprivation vs those of after sufficient sleep (***Figure 2b***). Together, these results demonstrate a significantly *lower* cortical inhibition and *higher* cortical facilitation after sleep deprivation.

## I-wave facilitation

In this double-pulse TMS protocol, cortical inhibition is reflected by I-wave peaks which are mainly observed at three ISIs occurring at 1.1–1.5 ms (early), 2.3–2.9 ms (middle), and 4.1–4.4 ms (late) after test pulse application. The results of the 2×3 ANOVA showed a significant interaction of sleep condition × ISI ($F_{1.74}$=14.59, p<0.001, $\eta p^2$=0.33), and main effects of sleep condition ($F_1$=20.36, p<0.001, $\eta p^2$=0.41) and ISI ($F_{1.67}$=47.39, p<0.001, $\eta p^2$=0.62) on I-wave peak MEP amplitudes. Bonferroni-corrected post hoc comparisons showed a significant increase of I-wave peaks for early and middle ISIs, vs single-pulse MEPs after both sleep conditions. The I-wave peaks for late ISI were significant only after sleep deprivation. Importantly, the peaks (at all ISIs) were significantly larger after sleep deprivation vs sufficient sleep (***Figure 2c***). These results indicate reduced GABAergic inhibition, resulting in I-wave facilitation, after sleep deprivation.

## Short-latency afferent inhibition

In this protocol, the TMS stimulus is coupled with peripheral nerve stimulation that has an inhibitory effect on motor cortex excitability at ISIs of 20 and 40 ms. Smaller MEPs indicate cortical inhibition. A significant interaction of sleep condition × ISI ($F_{1.81}$=27.51, p<0.001, $\eta p^2$=0.48) and a significant main effect of sleep condition ($F_1$=70.18, p<0.001, $\eta p^2$=0.71), but not ISI ($F_1$=1.58, p<0.217, $\eta p^2$=0.05) were observed on MEP amplitudes. Bonferroni-corrected post hoc comparisons revealed a significantly pronounced inhibitory effect of peripheral stimulation on cortical excitability after *sufficient sleep*, compared to the single TMS pulse at both ISIs. However, respective MEPs were significantly *converted* to excitatory effects after sleep deprivation. Moreover, cortical inhibition was significantly reduced after sleep deprivation vs sufficient sleep at the respective ISIs (***Figure 2d***). In line with the SICI and I-wave protocols, this suggests a reduction of cortical inhibition and its *conversion* to excitatory effects after sleep deprivation.

Taken together, our results demonstrate that glutamate-related intracortical excitability is upscaled after sleep deprivation. Moreover, cortical inhibition is decreased or turned into facilitation, which is indicative of enhanced cortical excitability as a result of GABAergic reduction. Corticospinal excitability did only show a trendwise upscaling, indicative for a major contribution of cortical, but not downstream excitability to this sleep deprivation-related enhancement. Cortical excitability is closely related to LTP/LTD plasticity in the brain and is expected to be related to changes of synaptic potentiation after lack of sleep (***Tononi and Cirelli, 2003***; ***Kuhn et al., 2016***). Accordingly, in the next step, we investigated the impact of non-invasively inducing LTP/LTD-like neuroplasticity under sleep deprivation vs sufficient sleep conditions.

**Table 1.** Mixed-model ANOVA results for the effect of tDCS on MEP amplitudes after sufficient sleep and sleep deprivation.

| Factor | df | F | p | $\eta p^2$ |
|---|---|---|---|---|
| Group | 1 | 11.74 | **0.002** | 0.296 |
| Sleep condition | 1 | 0.735 | 0.399 | 0.026 |
| Stimulation state | 1 | 37.09 | **<0.001** | 0.570 |
| Timepoint | 4.90 | 5.76 | **<0.001** | 0.171 |
| Sleep condition × group | 1 | 61.97 | **<0.001** | 0.689 |
| Stimulation state × group | 1 | 3.600 | 0.068 | 0.114 |
| Timepoint × group | 4.90 | 6.48 | **<0.001** | 0.188 |
| Sleep condition × timepoint | 5.18 | 0.445 | 0.822 | 0.016 |
| Stimulation state × timepoint | 5.08 | 2.91 | **0.015** | 0.094 |
| Sleep condition × stimulation state | 1 | 5.28 | **0.029** | 0.159 |
| Sleep condition × stimulation state × group | 1 | 57.99 | **<0.001** | 0.674 |
| Sleep condition × timepoint × group | 5.18 | 9.40 | **<0.001** | 0.251 |
| Stimulation state × timepoint × group | 5.08 | 3.68 | **0.003** | 0.116 |
| Sleep condition × stimulation state × timepoint | 5.35 | 1.09 | 0.369 | 0.037 |
| Group × sleep condition × stimulation state × timepoint | 5.35 | 12.14 | **<0.001** | 0.301 |

*Note:* tDCS: transcranial direct current stimulation; MEP: motor-evoked potentials. Significant effects are marked in **bold** (where p<0.05), n=30 (15 per group).

## Sleep deprivation saturates induction of LTP-like plasticity and converts the direction of LTD-like plasticity

Here, we were interested in determining how sleep deprivation and the resultant upscaled cortical excitability, affect LTP- and LTD-like plasticity in the brain. The sleep synaptic hypothesis proposes that synaptic strength is saturated during long awake times and restored after sleep (***Tononi and Cirelli, 2014***). Saturation can lead to decreased LTP-like plasticity in humans (***Kuhn et al., 2016***). Accordingly, we expected that induction of LTP-like plasticity is decreased due to saturated synaptic strength and hyperexcited brain state identified in the previous section. We were also interested in determining how induction of LTD-like plasticity is affected under this brain state, which has not been investigated so far. To this end, participants received 'anodal vs sham' and 'cathodal vs sham' tDCS over the motor cortex after sufficient sleep and sleep deprivation (see Methods). We analyzed the MEPs by a mixed-model ANOVA with stimulation condition (active, sham), timepoint (seven levels), and sleep condition (normal vs deprivation) as within-subject factors and group (anodal vs cathodal) as between-subject factor. A significant four-way interaction of sleep condition × group × tDCS state × timepoint was found ($F_{5.35}$=12.14, p<0.001, $\eta p^2$=0.30), indicating that tDCS-induced LTP/LTD-like neuroplasticity was differentially affected in the sleep conditions. Other interactions and main effects are summarized in *Table 1*. Baseline MEPs did not significantly differ across sleep and stimulation conditions (***Supplementary file 1C***). Reported side effects and analyses of blinding efficacy can be found in the supplementary material (***Supplementary file 1D and E***).

### Anodal LTP-like-induced plasticity

Bonferroni-corrected post hoc t-tests reveal that after sufficient sleep, anodal tDCS significantly increased MEP amplitudes immediately after 5, 10, 15, 20, and 25 min after the intervention. The increase of MEP amplitudes was significantly larger at all timepoints when compared to sleep deprivation and against the sham intervention (*Figure 3a and b*). In contrast, sleep deprivation prevented induction of LTP-like plasticity via anodal tDCS at all timepoints. No significant effect of anodal tDCS

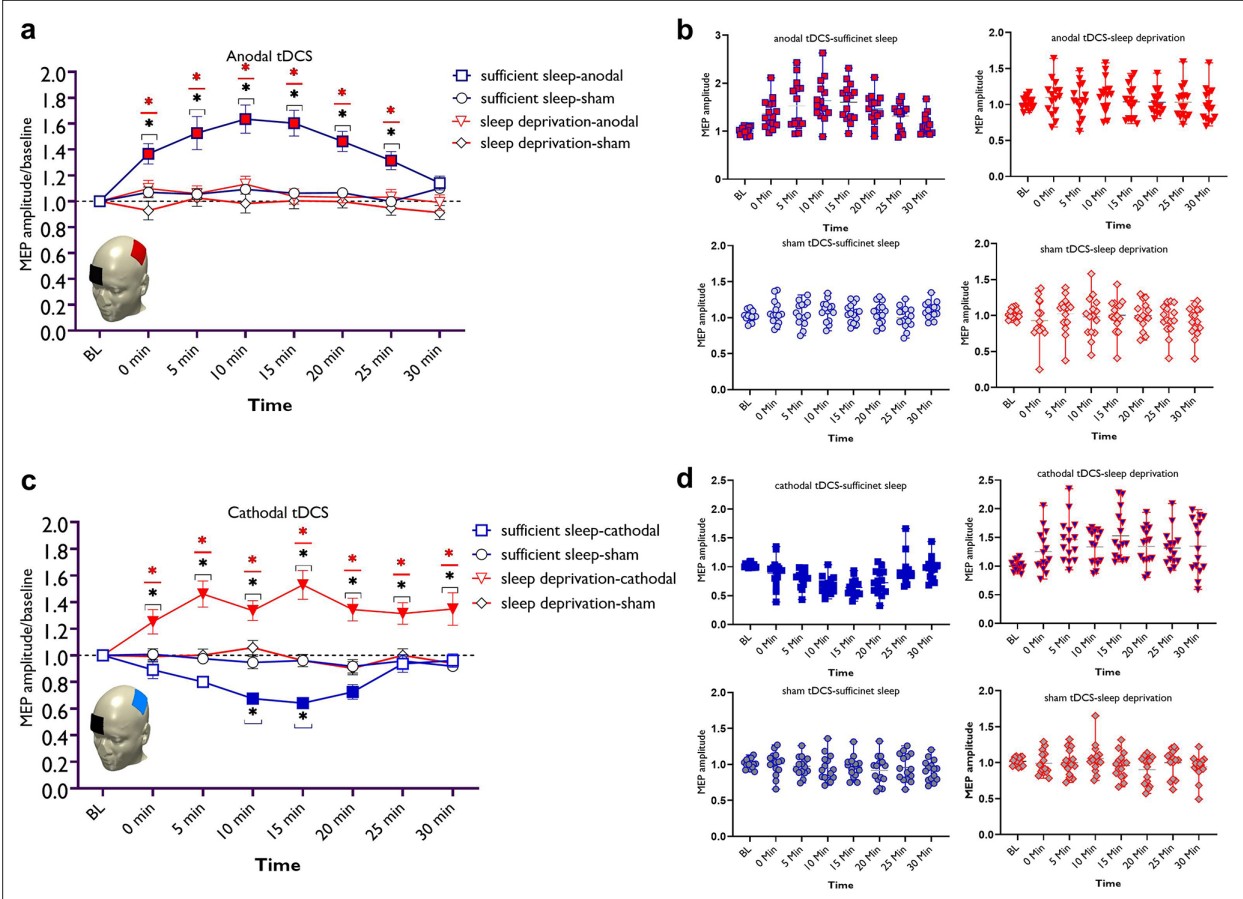

**Figure 3.** LTP/LTD-like plasticity induction after sufficient sleep vs sleep deprivation. (**a**) Cortical excitability alterations after inducing LTP-like plasticity with anodal transcranial direct current stimulation (tDCS) sleep conditions. Post hoc comparisons (Bonferroni-corrected) of motor-evoked potential (MEP) amplitudes to respective baseline values, the sham condition, and sleep conditions are marked by symbols in the figures. (**b**) Cortical excitability alterations after inducing LTD-like plasticity with cathodal tDCS under sleep conditions. Sham stimulation after both, sufficient sleep and sleep deprivation did not induce any significant change in cortical excitability. Filled symbols indicate a significant difference of cortical excitability against the respective baseline values. The black asterisks (*) indicate a significant difference between the real vs sham tDCS conditions, and the red asterisks (*) indicate a significant difference between respective timepoints of tDCS conditions after sufficient sleep vs sleep deprivation. All error bars represent the s.e.m. (**c,d**) Individual mean MEPs variability obtained from tDCS conditions after sufficient sleep and sleep deprivation. The X-axis represents timepoint (Bl, 0, 5, 10, 15, 20, 25, 30 min) and the Y-axis represents mean MEP amplitudes. n=30 (15 per group).

was observed when compared to the baseline and against the sham intervention after sleep deprivation for all timepoints.

## Cathodal LTD-like-induced plasticity

Here, post hoc analyses show that after sufficient sleep, LTD-like plasticity was induced (decreased MEP amplitudes) via cathodal tDCS at 10, 15, and 20 min timepoints compared to baseline MEP. The MEP amplitudes at 10 and 15 min timepoints were significantly different from respective timepoints in the sham condition. Importantly, the MEP decrease was significantly larger at all timepoints when compared to MEP size after sleep deprivation. Sleep deprivation, interestingly, *reversed* the inhibitory LTD-like aftereffects of cathodal stimulation into excitatory LTP-like aftereffects. Specifically, sleep deprivation led to an increase of MEP amplitudes (LTP-like) at all timepoints when compared to the baseline, against the sham intervention and compared to the respective timepoints after sufficient sleep (*Figure 3c and d*). This excitability-enhancing effect of cathodal stimulation was longer-lasting too, as shown by a sustained MEP amplitude enhancement at the 30 min timepoint when MEP amplitudes are expected to be back at baseline levels.

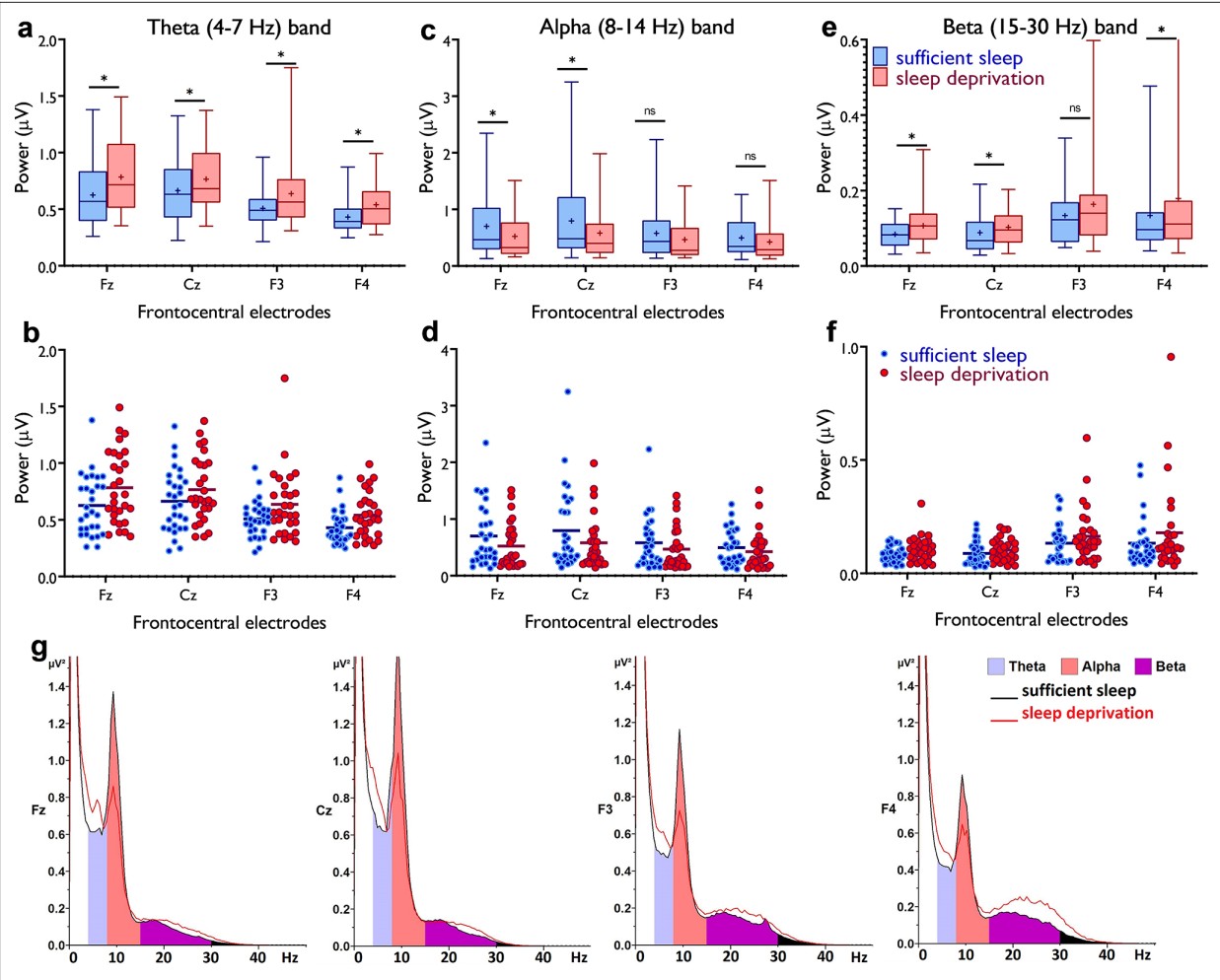

**Figure 4.** Resting-state theta, alpha, and beta oscillations at electrodes Fz, Cz, F3, and F4. (**a,b**) Theta band activity was significantly *higher* after the sleep deprivation vs sufficient sleep condition ($t_{Fz}$=4.61, p<0.001; $t_{Cz}$=2.22, p=0.034; $t_{F3}$=2.93, p=0.007; $t_{F4}$=4.78, p<0.001). (**c,d**) Alpha band activity was significantly *lower* at electrodes Fz and Cz ($t_{Fz}$=2.39, p=0.023; $t_{Cz}$=2.65, p=0.013) after the sleep deprivation vs the sufficient sleep condition. (**e,f**) Beta band activity was significantly *higher* at electrodes Fz, Cz, and F4 after sleep deprivation compared with the sufficient sleep condition ($t_{Fz}$=3.06, p=0.005; $t_{Cz}$=2.38, p=0.024; $t_{F4}$=2.25, p=0.032). (**g**) Power spectrum including theta (4–7 Hz), alpha (8–14 Hz), and beta (15–30 Hz) bands at the electrodes Fz, Cz, F3, and F4, respectively. Data of one participant were excluded due to excessive noise. All pairwise comparisons for each electrode were calculated via *post hoc* Student's t-tests (paired, p<0.05). n=29. Error bars represent s.e.m. ns: nonsignificant; asterisks (*) indicate significant differences. Boxes indicate the interquartile range that contains 50% of values (range from the 25th to the 75th percentile) and whiskers show the 1st–99th percentiles.

## Electrophysiological evidence of upscaled cortical excitability and saturated synaptic potentiation

So far, we found that sleep deprivation upscales cortical excitability, prevents induction of LTP-like plasticity, presumably due to saturated synaptic potentiation, and converts LTD- into LTP-like plasticity. We next investigated how sleep deprivation affects resting-state brain oscillations at the theta band (4–7 Hz) as indirect markers of homeostatic sleep pressure (*Vyazovskiy and Tobler, 2005*; *Leemburg et al., 2010*) and increased excitability (*Kuhn et al., 2016*), the beta band (15–30 Hz) as another marker of cortical excitability, vigilance, and arousal (*Eoh et al., 2005*; *Fischer et al., 2008*), and the alpha band (8–14 Hz) which is important for cognition (e.g. memory, attention) (*Klimesch, 2012*). To this end, we analyzed EEG spectral power at mid-frontocentral electrodes (Fz, Cz, F3, F4) using a 4×2 mixed ANOVA. For theta activity, significant main effects of location ($F_{1.71}$=18.68, p<0.001; $\eta p^2$=0.40) and sleep condition ($F_1$=17.82, p<0.001; $\eta p^2$=0.39), but no interaction was observed, indicating that theta oscillations at frontocentral regions were similarly affected by sleep deprivation. Post hoc tests (paired, p<0.05) revealed that theta oscillations, grand averaged at mid-central electrodes,

were significantly *increased* after sleep deprivation (p<0.001) (*Figure 4a and b*). For the alpha band, the main effects of location ($F_{1.49}$=12.92, p<0.001; $\eta p^2$=0.31) and sleep condition ($F_1$=5.03, p=0.033; $\eta p^2$=0.15) and their interaction ($F_{2.31}$=4.60, p=0.010; $\eta p^2$=0.14) were significant. Alpha oscillations, grand averaged at mid-frontocentral electrodes, were significantly *decreased* after sleep deprivation (p=0.033; *Figure 4c and d*). Finally, the analysis of beta spectral power showed significant main effects of location ($F_{1.34}$=6.73, p=0.008; $\eta p^2$=0.19) and sleep condition ($F_1$=6.98, p=0.013; $\eta p^2$=0.20) but no significant interaction. Beta oscillations, grand averaged at mid-frontocentral electrodes, were significantly *increased* after sleep deprivation (p=0.013; *Figure 4e and f*). These electrophysiological data support findings from the previous sections that sleep deprivation upscales cortical excitability and leads to synaptic saturation.

## Sleep deprivation compromises learning, memory formation, and cognitive performance

LTP and LTD are the primary mechanisms mediating learning and memory. Concentration of GABA (*Kolasinski et al., 2019*) and glutamate (*Stagg, 2014*) is important for motor learning and synaptic strengthening as well. Results of the resting-EEG data also showed decreased alpha activity which is critically involved in cognition (e.g. memory, attention; *Klimesch, 2012*). Showing these converging effects of sleep deprivation on brain physiology, we were interested in determining how sleep deprivation affects sequence learning and cognitive functions as behavioral indices of cortical excitability and neuroplasticity. To this end, we measured motor sequence learning using the serial reaction time task (SRTT), working memory with a three-back letter task, and attentional functioning with the Stroop and AX continuous performance test (AX-CPT). Electrophysiological correlates of task performance (e.g. ERP) were measured as well (see Methods).

### Motor sequence learning

The differences in the standardized reaction time (RT) of block 5 vs 6, indicative of learning *acquisition*, and block 6 vs 7, indicative of learning *retention*, were analyzed with a 3 (block) × 2 (sleep condition) repeated measures ANOVA. The results showed a significant interaction of block × sleep condition ($F_{1.95}$=7.03, p=0.002, $\eta p^2$=0.19) and the main effects of sleep condition ($F_1$=21.47, p<0.001, $\eta p^2$=0.42) and block ($F_{1.93}$=41.63, p<0.001, $\eta p^2$=0.58) as well. Post hoc comparisons revealed a significantly larger RT difference at blocks 6–5 and blocks 6–7 only after sufficient sleep and lower committed errors (*Figure 5a and b*). Absolute RT, error rate, and RT variability were similarly affected by sleep deprivation (*Figure 5—figure supplement 1*). Next, we explored electrophysiological correlates of motor learning. The P300 component is evoked in response to stimuli of low probability and stimulus sequence (*Squires et al., 1976*). We expected a higher-amplitude P300 component, when the learned sequence of stimuli is violated (at block 6), after having sufficient sleep. We analyzed the P300 amplitudes (250–500 ms) in blocks (5–7) and the results revealed only a significant main effect of block on the amplitude at electrodes Pz ($F_{1.78}$=15.88, p<0.001, $\eta p^2$=0.35) and P3 ($F_{1.90}$=6.63, p=0.003, $\eta p^2$=0.18), which are among regions of interest in this task. The P300 amplitude in block 6 vs blocks 5 and 7 was significantly larger at electrode Pz after both sleep conditions, but respective comparisons between blocks at electrode P3 were significant only after sufficient sleep (*Figures 5c and 6a*). A similar trend was observed for the other electrodes of interest (*Figure 5—figure supplement 2*).

### Working memory

For working memory performance, the ANOVA results revealed a significant main effect of sleep condition on the N-back hits ($F_1$=12.36, p<0.001; $\eta p^2$=0.30), and *d* prime ($F_1$=11.77, p=0.002; $\eta p^2$=0.278) as the primary outcomes of interest, but not on RT of hits ($F_1$=0.01, p=0.894). Post hoc analyses showed significantly enhanced WM performance with significantly more RT variability after sufficient sleep, which could be due to an accuracy-RT trade-off (*Figure 5e and f*). Furthermore, the P300 ERP component was investigated across sleep conditions. No significant main effect of sleep condition was observed on the P300 component at electrodes Fz ($F_1$=1.66, p=0.208), Pz ($F_1$=0.88, p=0.364), and Cz ($F_1$=1.01, p=0.310). Yet, a trendwise increase of the P300 amplitude was identified at electrodes Fz and Cz after sufficient sleep compared to sleep deprivation (*Figures 5f and 6b*).

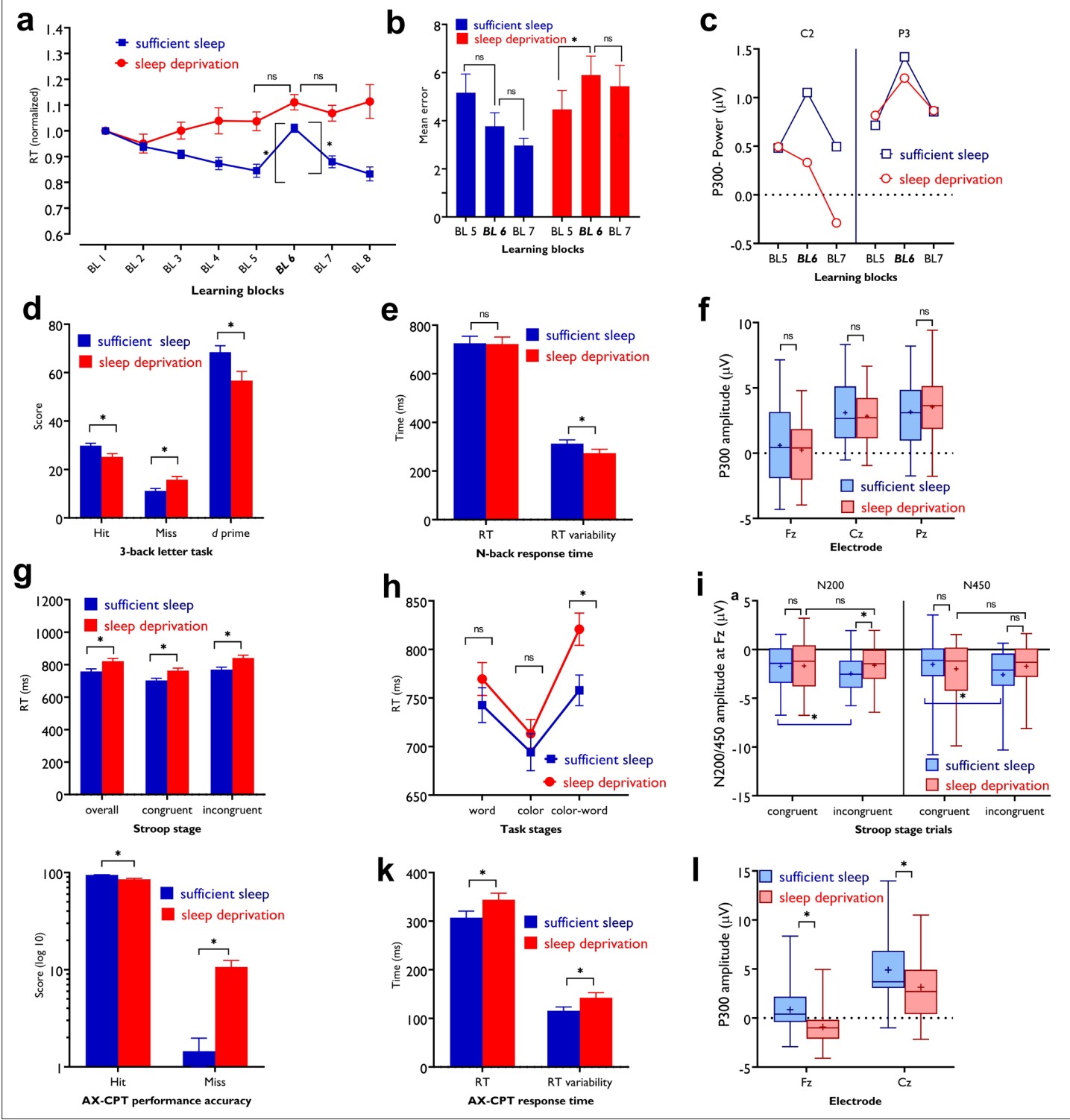

**Figure 5.** Impact of sleep deprivation on sequence learning, working memory, and attention. (**a**) The reaction time (RT) difference of block (BL) 6–5 (learning acquisition) and BL 6–7 (learning retention) was significant only after sufficient sleep (BL 6–5: $t$=3.73, p<0.001; BL 6–7: $t$=2.95, p=0.003) but not sleep deprivation (BL 6–5: $t$=1.67, p=0.094; BL 6–7: $t$=0.95, p=0.337).( **b**) Performance was more erroneous after sleep deprivation. Asterisks (*) represent significant differences between learning block RTs (BL 6–5, BL 6–7). n=30.(**c**) For both P3 and C2 electrodes, the P300 amplitude (250–500 ms) was significantly larger in block 6 vs BL5 and 7 only after sufficient sleep (P3: $t_{6-5}$=3.50, p<0.001, $t_{6-7}$=3.30, p=0.003; C2: $t_{6-5}$=2.74, p=0.010, $t_{6-7}$=2.64, p=0.013) marked by the filled symbol. n=30. (**d**) Participants had more correct responses ($t$=3.56, p<0.001) and a higher $d$ index ($t$=3.43, p=0.002) after having sufficient sleep vs sleep deprivation. (**e**) Performance speed was not significantly different but was more variable after sufficient sleep. n=30. (**f**) The P300

*Figure 5 continued on next page*

Figure 5 continued

amplitude (300–600 ms) did not significantly differ across sleep conditions at electrodes Fz, Pz, and Cz. n=29. (**g**) RT of the congruent, incongruent, and overall trials in the Stroop task was significantly slower after sleep deprivation. (**h**) Participants displayed a significantly stronger Stroop interference effect (RT$_{incongruent}$-RT$_{congruent}$) after sleep deprivation vs sufficient sleep (t=2.63, p=0.009). n=29. (**i**) The N200 (200–300 ms) at electrode Fz was significantly larger for incongruent trials, but not congruent trials, after sufficient sleep vs sleep deprivation. Both N200 and N450 (400–550 ms) were significantly larger for incongruent vs congruent trials only after sufficient sleep. (**j,k**) Participants were less accurate in identifying AX trials (t=5.30, p<0.001), had slower RT (t=3.29, p=0.003), and showed a larger variability of RT (t=3.13, p=0.004) after sleep deprivation vs sufficient sleep. (**l**) The P300 (300–600 ms) amplitude was significantly larger after sufficient sleep at electrodes Fz and Cz. n=27. All pairwise comparisons were calculated via post hoc Student's t-tests (paired, p<0.05). Error bars represent s.e.m. ns: nonsignificant; Asterisks (*) indicate significant differences. Boxes indicate interquartile range that contains 50% of values (25th–75th) and whiskers show 1st–99th percentiles. See also Figure S1-S3.

The online version of this article includes the following figure supplement(s) for figure 5:

**Figure supplement 1.** The impact of sleep deprivation on motor learning performance.

**Figure supplement 2.** P300 amplitudes of electrodes C1, C2, P1, and P2 during motor sequence learning across sleep conditions.

**Figure supplement 3.** The impact of sleep deprivation on Stroop accuracy.

## Selective attention

The RT difference of congruent and incongruent trials in the Stroop task was analyzed with a 2 (sleep condition) × 2 (congruency) factorial ANOVA. Sleep condition ($F_1$=23.77, p<0.001; $\eta$p$^2$=0.45) and congruency ($F_1$=106.15, p<0.001; $\eta$p$^2$=0.78) had significant effects on Stroop interference but they did not interact ($F_1$=0.69, p=0.413). Post hoc comparisons revealed a significant Stroop effect (slower RT of incongruent trials vs congruent trials) after both, sufficient sleep (t=3.01, p=0.009) and sleep deprivation (t=3.47, p<0.001). However, the interference effect was significantly stronger after sleep deprivation vs sufficient sleep in overall trials (t=2.82, p=0.015), congruent trials (t=2.71, p=0.021), and incongruent trials (t=3.19, p=0.005). The number of committed errors in all stages of the task was significantly higher under the sleep deprivation condition (***Figure 5—figure supplement 3a***). Reduced Stroop effects are associated with higher N200 and N450 amplitudes, which are indicative of higher selective attention and better detection of conflicting stimuli. We analyzed these ERP components too. The results of the 2 (congruency) × 2 (sleep condition) ANOVA showed a significant interaction of sleep condition × congruency on the N200 ($F_1$=3.90, p=0.05; $\eta$p$^2$=0.12) and N450 amplitudes ($F_1$=6.43, p=0.017; $\eta$p$^2$=0.19) for the electrode Fz. The main effect of sleep condition was not significant, and the main effect of congruency was significant only for N450 component ($F_1$=4.29, p=0.045; $\eta$p$^2$=0.14). Sleep deprivation was related to a significantly *smaller* N200 amplitude, but not N450, for the incongruent trials only, at electrode Fz (t=2.75, p=0.010). Both N200 and N450 amplitudes of incongruent trials were significantly *larger* compared to congruent trials after sufficient sleep (t$_{N200}$=2.51, p=0.018; t$_{N450}$=3.63, p=0.001), but not sleep deprivation (t$_{N200}$=0.24, p=0.810; t$_{N450}$=0.48, p=0.634), indicating that conflict detection was more clearly processed after having sufficient sleep (***Figures 5i and 6c***). Results of the electrode Cz can be found in supplementary materials (***Figure 5—figure supplement 3b***).

## Sustained attention

The AX-CPT was used for measuring sustained attention. We found a significant main effect of sleep condition on performance accuracy ($F_1$=28.12, p<0.001; $\eta$p$^2$=0.49) as the primary outcome of interest, RT of hit trials ($F_1$=10.85, p=0.003; $\eta$p$^2$=0.27), and variability of RT ($F_1$=9.85, p=0.004; $\eta$p$^2$=0.25). Participants responded significantly less accurately, with slower RT, and more variable RT after sleep deprivation compared to sufficient sleep (***Figure 5m and n***). Here again, the P300 serves as an attentional index of the target stimulus and memory storage. Analysis of this ERP component showed a significant main effect of the sleep condition on the P300 component at electrodes Fz ($F_1$=20.25, p<0.001; $\eta$p$^2$=0.43), Cz ($F_1$=20.57, p<0.001; $\eta$p$^2$=0.44), but not Pz ($F_1$=0.72, p=0.402). Post hoc analyses indicated that sleep deprivation was related to a significantly smaller P300 amplitude in these (***Figures 5l and 6d***), and other electrodes of interest (***Figure 5—figure supplement 3c***).

## Relevant correlations

Although our study was not sufficiently powered for conducting correlational analyses between measures as primary outcome, we ran exploratory correlation analyses to identify associations between

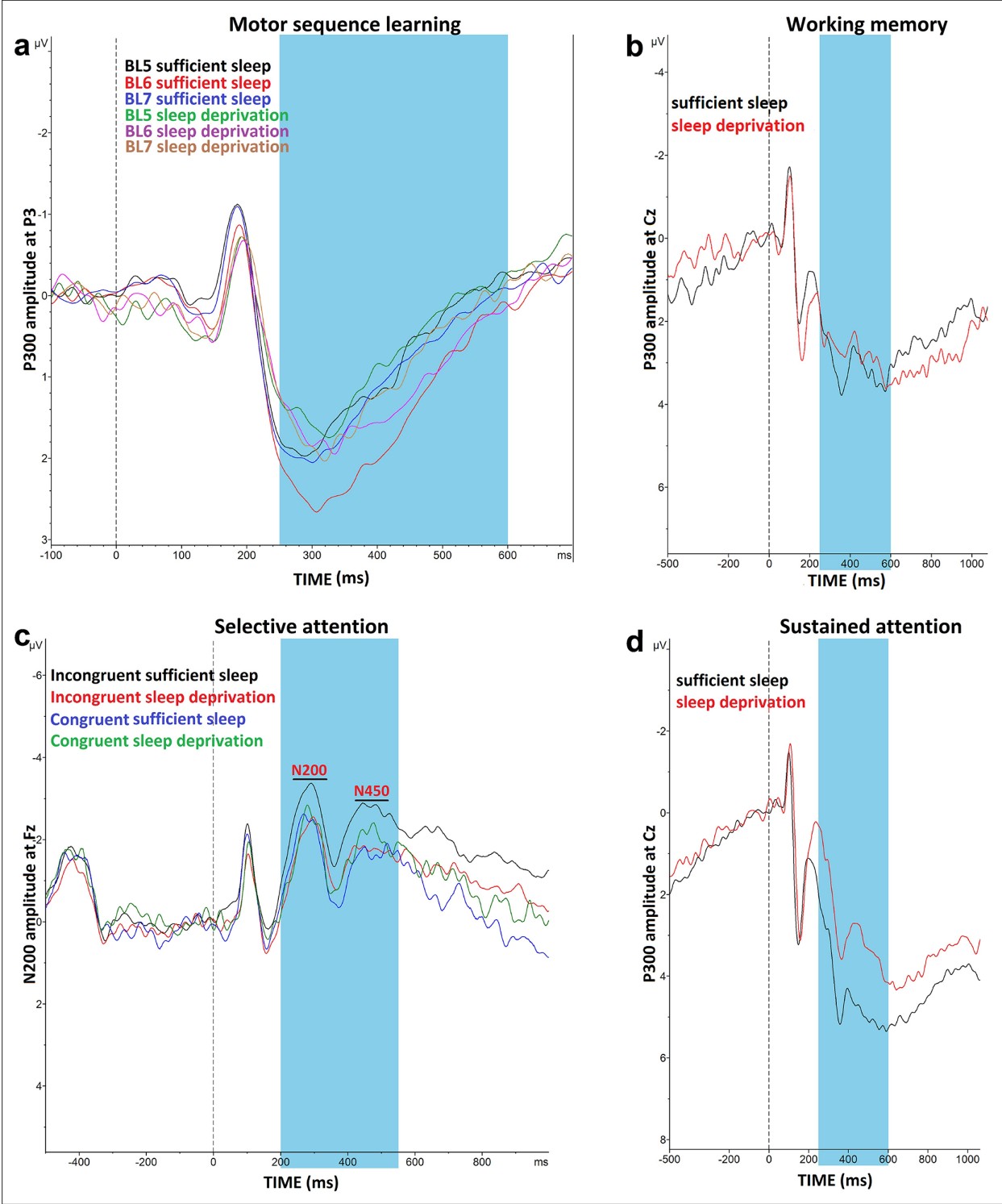

**Figure 6.** Impact of sleep deprivation on ERP components of sequence learning, working memory, and attention. (**a**) P300 amplitude (250–500 ms) at electrode P3 is significantly larger in block 6 vs blocks 5 and 7 only after sufficient sleep during motor sequence learning task. (**b**) P300 amplitude (300–600 ms) at electrode Cz for the n-back hits. (**C**) N200 (200–300 ms) and N450 (400–550 ms) amplitude at electrode Fz during selective attention task is significantly larger after sufficient sleep. (**d**) P300 amplitude (300–600 ms) during sustained attention task is significantly larger for hit trials at electrode Cz after sufficient sleep. ms: milliseconds. *Note*: In working memory ERP analysis (n=1), Stroop behavioral (n=1) ERP analyses (n=2), and AX-CPT ERP analyses (n=3), the data of some participants were excluded from the analysis due to excessive noise.

physiologically and cognitive parameters which are conceptually related, including measures of plasticity and motor learning on the one hand, and parameters of cortical excitability, working memory, and attention on the other. We found several relevant correlations between behavioral learning vs plasticity and cognition vs excitability indices. LTP-like plasticity effects after sufficient sleep were correlated with better sequence learning acquisition ($r=-0.558$, $p=0.031$) and retention ($r=-0.734$, $p=0.002$). Enhanced working memory and sustained attention after having sufficient sleep were also correlated with higher cortical facilitation and lower cortical inhibition (supplementary materials).

### Demographics, subjective sleepiness, and cortisol level

The mean age of participants was $24.62 \pm 4.16$ years (50% males). Age and gender did not correlate with the dependent variables discussed in the previous sections. Ratings of sleepiness and alertness at 9:00 AM showed a significantly higher sleep pressure, as measured by the Karolinska and Stanford Sleepiness Scales (KSS, SSS) after sleep deprivation (mean$_{KSS}$ = $7.10 \pm 1.76$, mean$_{SSS}$ = $4.90 \pm 1.32$), compared to the sufficient sleep condition (mean$_{KSS}$ = $2.96 \pm 0.764$, mean$_{SSS}$ = $2.23 \pm 0.62$). Sleep condition had a significant effect on KSS ($F_1$=159.02, $p<0.001$; $\eta p^2$=0.84) and SSS ($F_1$=122.10, $p<0.001$; $\eta p^2$=0.81) ratings, and a significantly higher sleep pressure was observed after the sleep deprivation vs the sufficient sleep. The average levels of cortisol and melatonin were numerically lower after sleep deprivation vs sufficient sleep (cortisol: $3.51 \pm 2.20$ vs $4.85 \pm 3.23$, $p=0.056$; melatonin: $10.50 \pm 10.66$ vs $16.07 \pm 14.94$, $p=0.16$), but these differences were only marginally significant for the cortisol level and showed only a trendwise reduction for melatonin.

## Discussion

In this study, we investigated how cortical excitability, brain stimulation-induced neuroplasticity, and cognitive functions are affected by one-night sleep deprivation. We hypothesized an increase and decrease in intracortical facilitation and inhibition, respectively, lower neuroplasticity induction for both LTP and LTD, and compromised behavioral performance in learning, memory, and attention tasks. In line with our hypotheses and recent previous works, sleep deprivation upscaled parameters of cortical excitability, and corticospinal excitability was trendwise upscaled. This was associated with diminished induction of LTP-like plasticity under sleep pressure. The induction of LTD-like plasticity with cathodal tDCS, however, and in contrast to our initial hypothesis, was converted into LTP-like plasticity (anodal like) after sleep deprivation. These sleep deprivation-related physiological changes in the human brain provide further support for the synaptic homeostasis hypothesis, according to which sleep plays a critical role in desaturating synaptic potentiation (*Tononi and Cirelli, 2014*; *de Vivo et al., 2017*) and adds important novel insights into how LTD-like plasticity is affected under sleep pressure. These physiological findings were moreover associated with compromised sequence learning, working memory, and attentional functioning and their electrophysiological correlates, which underscore the behavioral relevance of synaptic homeostasis. Finally, changes in concentration of cortisol and melatonin were minor at best, in line with recent works (e.g. *Kuhn et al., 2016*) and cannot explain the observed effects.

Changes in cortical excitability following sleep deprivation were mostly observed in the corticocortical measures that include neurotransmitter systems involved in intracortical facilitation and inhibition (*Chen, 2000*; *Di Lazzaro et al., 2000*; *Di Lazzaro et al., 2005a*), and brain oscillations. Specifically, glutamate-related cortical facilitation (measured by ICF) was upscaled while GABA- and acetylcholine-related cortical inhibition (measured by SICI, I-wave, and SAI) were disinhibited or reversed after sleep deprivation. These sleep-dependent changes of cortical excitability are in line with the synaptic homeostasis hypothesis postulating that synaptic strength is increasingly potentiated during wakefulness and saturated if wakefulness is extended (*Tononi and Cirelli, 2003*; *Vyazovskiy et al., 2008*). Animal studies have shown that at this state, molecular and electrophysiological markers of synaptic strength *increase* including AMPA receptors, cortical spine density, slope and amplitude of cortical evoked responses, and even the size and number of synapses (*Tononi and Cirelli, 2003*; *Vyazovskiy et al., 2008*; *Bushey et al., 2011*; *Maret et al., 2011*). Our findings also complement those of human studies that show an increase and decrease of ICF and inhibition, respectively (*Huber et al., 2013*; *Kuhn et al., 2016*), as well as those animal and human studies that documented an increase of theta and beta band activity, markers of homeostatic sleep pressure, and after sleep deprivation (*Finelli*

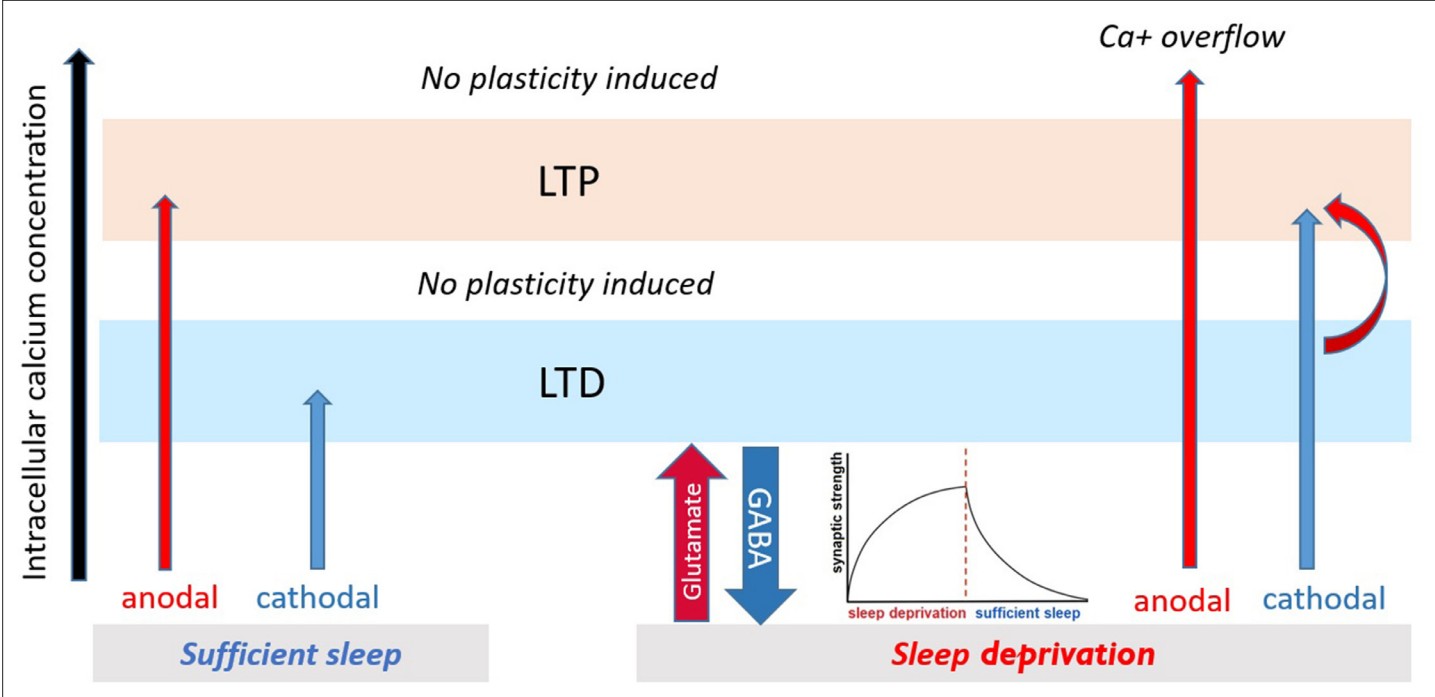

**Figure 7.** Proposed mechanism for plasticity induction. The intracellular calcium concentration (x-axis) determines directionality of plasticity (*Lisman, 2001*). It can be assumed that intracellular calcium concentration under no sleep pressure (sufficient sleep) is at an optimal level leading to stronger tDCS-induced LTP/LTD-like plasticity. Under sleep deprivation, LTP-like plasticity induction via anodal tDCS is prevented due to calcium overflow, and LTD-like plasticity via cathodal tDCS is converted to LTP-like plasticity possibly via (**a**) enhanced baseline calcium (due to upscaled excitability) which makes minor calcium increase obtained from cathodal stimulation to be sufficient to induce LTP-like plasticity and (**b**) gradual downregulating upscaled cortical excitability and opening some synaptic space for LTP-related plasticity induction.

*et al., 2000*; *Vyazovskiy and Tobler, 2005*; *Leemburg et al., 2010*; *Kuhn et al., 2016*). Together, these findings argue for an upscaled state of cortical excitability mediated by the need for sleep.

Cortical excitability is a basic physiological response of cortical neurons to an input and is, therefore, a fundamental aspect of neuroplasticity and cognition (*Kuhn et al., 2016*; *Ly et al., 2016*; *Gaggioni et al., 2019*). Changes in brain excitability will therefore affect neuroplasticity and cognition. We non-invasively induced LTP- and LTD-like plasticity by tDCS to investigate how a synaptic saturation state due to sleep deprivation affects the inducibility of LTP- and LTD-like plasticity. For LTP-like plasticity, the results demonstrate diminished plasticity in the motor cortex after sleep deprivation as compared to LTP-like plasticity induction after sufficient sleep. This can be explained via the synaptic strength perspective. Sleep deprivation saturates (or upscales) synaptic strength which eventually leads to deficient LTP inducibility. This is moreover in agreement with in vivo and in vitro studies in rats (*McDermott et al., 2003*; *Kopp et al., 2006*; *Vyazovskiy et al., 2008*), and in line with the results of a recent human study that showed decreased LTP-like plasticity after sleep deprivation (*Kuhn et al., 2016*). Regarding LTD, the results show that sleep deprivation *reversed* the LTD-like inhibitory effect of cathodal stimulation into LTP-like excitatory effects (i.e., anodal-like). This is the first evidence for the effect of sleep deprivation on LTD-like plasticity induction in humans. Critically, this conversion of LTD-like into LTP-like plasticity is in line with a saturating effect of sleep deprivation on synaptic strength rather than cortical underactivation, which would be otherwise an alternative explanation for reduced LTP-like effects after anodal tDCS.

One mechanism that can explain the diminished LTP-like and conversion of LTD-like to LTP-like plasticity during sleep deprivation is intracellular calcium concentration (*Figure 7*). It is known that the directionality of plasticity (LTP or LTD) depends on the level of calcium concentration (*Lisman, 2001*) with LTP and LTD being linked to higher and lower intracellular $Ca^{2+}$ concentration, respectively. There are, however, certain limit zones for induction of plasticity. In this line, anodal LTP-like plasticity is linked to largely enhanced intracellular calcium concentration in animals and humans (*Islam et al., 1995*; *Nitsche et al., 2005*; *Biabani et al., 2018*), while cathodal tDCS-induced LTD-like plasticity

is assumed linked to lower intracellular $Ca^{2+}$ concentration (*Nitsche et al., 2003b*). Our excitability results earlier showed a glutamate-controlled NMDA receptor facilitation during sleep deprivation, which enhances calcium influx. In this state, applying anodal tDCS, which enhances calcium concentration, can then result in abolishment of LTP induction due to calcium overflow (*Lisman, 2001*; *Misonou et al., 2004*; *Grundey et al., 2018*). On the other hand, inducing LTD-like plasticity might downregulate upscaled cortical excitability at the network level and open some synaptic space for LTP-related plasticity induction. This is in line with previous studies that show calcium enhancement can convert the direction of cathodal-induced LTD-like to LTP-like plasticity (*Batsikadze et al., 2013*).

We also monitored EEG oscillatory activities and found an increase of theta-band activity in the frontal-midline regions during sleep deprivation. These oscillatory activities constitute the most salient electrophysiological correlate of sleep pressure-related brain activity. There are, however, also other potentially interesting components of sleep-related theta activity and also potentially sleep deprivation-related theta activity in the human EEG. It was recently shown that synchronized theta oscillations (i.e. periodic) during sleep occur in transient bursts that are interleaved with desynchronized aperiodic network states that represent information-rich neurophysiological substrates of sleep-dependent cognition-related mechanisms, including memory formation and synaptic homeostasis (*Helfrich et al., 2021*). These aperiodic components are involved in (1) hippocampal–neocortical network interactions, (2) excitation–inhibition (E/I) balance, and (3) plasticity (*Hanslmayr et al., 2016*; *Helfrich et al., 2021*). Our main objective in this study was to analyze EEG markers of sleep pressure, and not sleep, and thus those EEG markers of sleep (e.g. hippocampus-dependent theta oscillations) along with related aperiodic components of EEG activity were beyond the scope of this study. These analyses were thus not carried out here, however, they may provide new information about functional EEG markers during sleep deprivation related to plasticity and the capacity of information processing (*Hanslmayr et al., 2016*; *Helfrich et al., 2021*).

Changes in synaptic strength are the primary mechanisms mediating learning and memory (*Feldman, 2009*). The sleep-dependent global synaptic downscaling spares neuronal assemblies crucially involved in the encoding of information (*Niethard and Born, 2019*) which results in an improved signal-to-noise ratio and a renewed capacity for encoding new information (*Kuhn et al., 2016*). Accordingly, a better cognitive/behavioral performance is expected after sleep, compared to sleep deprivation. In this line, our results show enhanced motor sequence learning after sufficient sleep and impaired sequence learning and retention after sleep deprivation. Similarly, we show compromised working memory and attentional functioning after sleep deprivation in line with previous findings across species (*Havekes et al., 2016*; *Zare Khormizi et al., 2019*). In support of this, we noticed specific alterations of task-dependent ERP components, such as lower amplitudes of the P300 after sleep deprivation in the sustained attention, and motor learning tasks, and suppressed N200 and N450 in the Stroop conflict condition, which measures selective attention and interference control.

To converge our findings, the potential links between the physiological parameters of the brain and behavioral tasks are noteworthy. In sequence learning, a specific link can be made between behavioral learning and plasticity results. The behavioral and electrophysiological markers of motor sequence learning and retention were associated with facilitated LTP/LTD-like plasticity in the motor cortex after sufficient sleep, which supports the suggested link between inducibility of neuroplasticity and learning and memory formation. This observation makes sense, as tDCS-induced neuroplasticity in the motor cortex and behavioral motor learning share intracortical mechanisms (*Stagg, 2014*). For the cognitive functions measured in our study, cortical excitability alterations are more relevant. The increase of cortical excitability parameters and the resultant synaptic saturation following sleep deprivation can explain the respective cognitive performance decline. It is, however, worth noting that our study was not powered to identify these correlations with sufficient reliability and future studies that are powered for this aim are needed.

Our findings have several implications. First, they show that sleep and circadian preference (i.e. chronotype) have functionally different impacts on human brain physiology and cognition. The same parameters of brain physiology and cognition were recently investigated at circadian optimal vs non-optimal time of day in two groups of early and late chronotypes (*Salehinejad et al., 2021*). While we found decreased cortical facilitation and lower neuroplasticity induction (same for both LTP and LTD) at the circadian nonpreferred time in that study (*Salehinejad et al., 2021*), here we observed upscaled cortical excitability and a functionally different pattern of neuroplasticity alteration (i.e. diminished

LTP-like plasticity induction and conversion of LTD- to LTP-like plasticity). Second, the results of the present study underscore the importance of sufficient sleep for adaptive and efficient performance (e.g. in working or educational environments) specifically in individuals with demanding jobs. Third, the relevance of sleep for brain plasticity and optimal cortical excitability argues for the role of sufficient sleep in preventing disorders that are associated with cognitive and/or plasticity deficits, such as Alzheimer's disease (*Ju et al., 2014*) and major depression (*Wolf et al., 2016*) that constitute a growing public health concern. Aging, which is associated with cognitive decline, is another critical period of life that is associated with sleep disturbances (*Lim et al., 2013*). Furthermore, the findings of cortical excitability and neuroplasticity results have implications for the variability of NIBS, especially TMS and tDCS (*Polanía et al., 2018*). Controlling for sleep pressure should be considered in basic and applied studies that use NIBS. Finally, modulating sleep via therapeutic sleep deprivation (*Wolf et al., 2016*) or NIBS (*Romanella et al., 2020*; *Herrero Babiloni et al., 2021*) and thereby affecting plasticity and cognitive parameters seems a promising yet understudied field for the future.

There are several points to be considered for interpreting the results. First, timing of the experimental measurements should be taken into account. Although we strictly controlled for external and environmental factors that could affect cortical excitability, especially in the sleep-wake cycle, the measurement time of cognitive functions, cortical excitability monitoring, and neuroplasticity induction differed by a few hours (24, 26, 27 hr sleep deprivation, respectively). By fixing the specific time for conduction of the single protocols, we, however, controlled for potential confounding effects of different measurement times *within* a specific protocol. Nevertheless, for the between-protocol comparisons, this procedure could have gradually affected subjective sleepiness and probably brain physiology (*Kuhn et al., 2016*). Second, the physiological measures were based on the motor system and indirect measures of the involved neurotransmitters, and the cognitive tasks under study are more closely related to prefrontal regions, with the exception of the motor learning task. Third, the power spectral analysis was analyzed with a 'band ratio' approach in which each frequency band is analyzed based on periodic, or oscillatory, activity. However, the EEG signal reflects at least some aperiodic components (e.g. power spectrum density; *Donoghue et al., 2020*) that were not taken into account, as the focus of our study was on sleep pressure which refers to periodic activity. Considering aperiodic EEG signals could, however, be relevant for physiological interpretations of sleep-related data in future studies. Finally, sleep consists of different stages including REM, non-REM, slow-wave sleep, and spindles that can have different impacts on plasticity which are not taken into account in our data.

In conclusion, this experiment provides further evidence in humans supportive of the sleep homeostasis hypothesis. General information of upscaled brain excitability, as shown in previous works, was complemented by evidence that shows alterations in specific parameters of cortical excitability (e.g. increased facilitation, decreased and/or converted inhibition). The saturated state for inducing LTP-like plasticity aligns with the increased synaptic strength after sleep deprivation. The results of the present study also provide first evidence for LTD-like plasticity being converted into LTP under sleep pressure in the human brain, in conceptual accordance with an hyperexcitable state under sleep deprivation. These findings complement current knowledge about the critical role of sleep for neuroplasticity and cognition in humans.

# Materials and methods
## Participants

Thirty healthy adult volunteers (15 females, mean age=24.44 ± 3.93) who met the inclusion criteria were recruited from the TU Dortmund University, Ruhr-University Bochum, and the surrounding community. Power analysis showed that for a medium effect size (*Minarik et al., 2016*; $f$=0.35 equivalent to partial eta squared=0.10), a minimum of 30 subjects are required to achieve 95% power at an alpha of 0.05 for the primary applied statistical test. All participants were right-handed non-smokers, with a regular sleep-wake pattern (determined by sleep diary) and underwent a medical screening to verify no history of neurological diseases, epilepsy or seizures, central nervous system-acting medication, metal implants, and current pregnancy. As gender and age may affect the sleep-wake cycle and brain excitability, we balanced participants' gender and kept the age range to early adulthood. Each participant took part in a test TMS session to become acquainted with experiencing stimulation and the study protocol. Female subjects were not examined during the menstrual period to ensure

hormonal changes would not interfere with the measurements. This study conformed to the Declaration of Helsinki guidelines and was approved by the Institutional Review Board (ethics code: 99). Participants gave informed consent and received monetary compensation.

## Study design and course of study

This was a randomized, sham-controlled, crossover study. Participants completed a sleep diary 2 weeks before the beginning of the experiment and attended two experimental sessions after having 'sufficient sleep (23:00-8:00)' and 'sleep deprivation (23:00-8:00)' overnight in counterbalanced order. The experimental sessions took place at a fixed time in the morning (9:00 AM). The interval between the sessions was at least 2 weeks. In each session, participants underwent the same experimental protocol. Each experimental session began with saliva sampling at 8:45 AM, followed by behavioral and cognitive task performance during EEG recording (1.5 hr), followed by cortical excitability monitoring sessions (1 hr), and finally followed by neuroplasticity induction with active and sham tDCS (1.5 hr). The reason for starting the session with behavioral/EEG measures was to take advantage of 1 hr between 8 and 9 AM for preparing the EEG cap, which took about 45 min on average. The order of measurements (behavioral/EEG, cortical excitability, tDCS) was identical across participants, to keep the start time of each measurement and the number of hours deprived from sleep fixed for each measure.

## Sleep conditions

In the 'sufficient sleep' condition, participants had to go to bed in their home environment at around 23:00 and have at least 8 hr of uninterrupted sleep. This was to prevent potential poor sleep in a new environment (i.e. the laboratory). The experiment was scheduled to start at 9:00 AM. Participants were refrained from drinking alcohol and coffee 12 hr before sleep time and afterward until the end of the session. In the case of poor sleep quality for any reason (measured by sleepiness rating scales) or unregular sleep pattern (sleep onset, wake-up time) informed by the sleep diary (more than ±2.5 hr deviation from the scheduled time frames), the respective session was canceled and postponed until sufficient sleep condition requirements were met. In the 'sleep deprivation' condition, participants spent the night in a specific lounge at the local institute where the data was collected. The lounge was equipped with an unrecording live camera and prepared for participants' stay overnight. Participants spent all night awake in the lounge (23:00–8:00) and were supervised by a scientific staff member. Additionally, their sleep-waking status was recorded via a wrist-worn Actigraphy (MotionWatch 8.0, CamTech, Cambridge, UK). Food and drinks were provided (the consumption of coffee, caffeine-containing soft drinks, black tea, and alcohol was not allowed), watching TV programs, reading, and working on the computer were also allowed. Participants were prevented from any sleep-related activities during the night, such as lying down, or closing their eyes for a prolonged time. They were refrained from drinking caffeine-containing drinks and sleeping or taking a nap from the afternoon before they joined the sleep deprivation session at 23:00. Before starting each experimental session, subjective sleepiness of participants and their alertness was evaluated with KSS (*Akerstedt and Gillberg, 1990*) and SSS (*Hoddes et al., 1972*). All external factors that could affect circadian rhythmicity such as light and food intake were controlled during the experiment.

## Determination of cortisol and melatonin from saliva

Saliva samples were collected using salivettes (Sarstedt AG & Co. KG, Germany). After centrifugation at 3000× g for 2 min at 4°C, samples were aliquoted at 500 µL each and stored at –20°C until measurement. Cortisol and melatonin were measured using Cortisol Saliva ELISA and Melatonin direct Saliva ELISA (both IBL International GmbH, Germany) according to manufacturer's instructions. Duplicate measurements were conducted for all samples in a range from 0.05 µg/mL to 30 µg/mL for cortisol using 50 µL of sample and from 1.0 pg/mL to 50 pg/mL for melatonin using 100 µL of sample. In case of values exceeding the measurement range for melatonin samples were diluted 1:10 and remeasured.

## Cortical excitability

Different protocols of single-pulse and paired-pulse TMS were used to monitor corticospinal and intracortical excitability in the motor cortex. These protocols included: resting motor threshold (RMT), active motor threshold (AMT), I-O curve, short intracortical inhibition and facilitation (SICI-ICF),

intracortical I-wave facilitation, and SAI. RMT, AMT, and I-O curve examine corticospinal excitability, SICI-ICF measures both, ICF and inhibition, and intracortical I-wave facilitation and SAI are measures of intracortical inhibition of the human motor cortex (*Kujirai et al., 1993*; *Chen, 2000*; *Di Lazzaro et al., 2000*).

### Single-pulse MEP, resting and active motor threshold

Single-pulse biphasic TMS at 0.25 Hz ± 10% (random) was through a figure-of-eight magnetic coil (diameter of one winding, 70 mm; peak magnetic field, 2T) held 45° to the midline and applied over the left primary motor cortex. Surface MEPs were recorded from the right abductor digiti minimi muscle (ADM) with gold cup electrodes in a belly-tendon montage. RMT was examined using the TMS Motor Threshold Assessment Tool (MTAT 2.0, http://www.clinicalresearcher.org/software.htm) (*Awiszus, 2003*) and was determined as the lowest stimulator intensity required to evoke a peak-to-peak MEP of 50 µV in the relaxed ADM muscle in at least 5 out of 10 consecutive trials. The AMT was determined as the lowest stimulator intensity required to elicit MEP response of ~200–300 µV during moderate tonic contraction of the right ADM muscle (~20% of the maximum muscle strength; *Rothwell et al., 1999*) in at least three of six consecutive trials.

### Input-output curve

The I-O curve is a TMS single-pulse protocol that reflects excitability of corticospinal neurons. It is modulated by glutamatergic activity and refers to the increase of MEP amplitudes with increasing TMS intensity (*Chen, 2000*). The slope of the recruitment curve increases at higher TMS intensities with higher glutamatergic and adrenergic transmission and decreases by drugs that enhance effects of GABA (*Chen, 2000*; *Paulus et al., 2008*). In the I-O curve protocol, MEP amplitudes in the relaxed right ADM muscle were measured in four blocks with different stimulus intensities (100, 110, 130, and 150% RMT; *Batsikadze et al., 2013*), each block with 15 pulses, and a mean (MEP amplitudes) was calculated for each intensity.

### Short-latency intracortical inhibition and intracortical facilitation

The SICI-ICF is a TMS paired-pulse protocol for monitoring of GABAergic-mediated cortical inhibition and the glutamate-mediated cortical facilitation (*Chen, 2000*). In this protocol, a subthreshold conditioning stimulus (determined as 70% of AMT) is followed by a suprathreshold test stimulus which was adjusted to evoke a baseline MEP of ~1 mV. The paired stimuli are presented in ISIs of 2, 3, 5, 10, and 15 ms (*Kujirai et al., 1993*). ISIs of 2 and 3 ms represent SICI and have inhibitory effects on test pulse MEP amplitudes, and ISIs of 10 and 15 ms represent ICF and have enhancing effects on single-pulse TMS-elicited MEP amplitudes (*Kujirai et al., 1993*; *Di Lazzaro et al., 1998*; *Di Lazzaro et al., 2003*). The stimuli (subthreshold and suprathreshold stimuli) were organized in blocks in which each ISI and one single test stimulus were applied once in pseudorandomized order. Each block was repeated 15 times, which resulted in a total of 90 single-pulse or paired-pulse MEP per session. The exact interval between the paired pulses was randomized (4 ± 0.4 s).

### Short-interval intracortical I-wave facilitation

This TMS protocol is based on I (indirect) waves which refer to high-frequency repetitive discharges of corticospinal neurons produced by single-pulse stimulation of the motor cortex (*Di Lazzaro et al., 2012*); for a detailed review see *Ziemann et al., 1998*; *Di Lazzaro et al., 2012*. In this protocol, two successive stimuli (supra- and subthreshold) are separated by short ISIs, but this protocol involves a suprathreshold first stimulus and a subthreshold second stimulus (*Ziemann et al., 1998*). The ISIs range from 1.1 ms to 4.5 ms latency and are presented in pseudorandomized order. We grouped ISIs to early (mean MEP at ISIs 1.1, 1.3, and 1.5 ms), middle (mean MEP at ISIs 2.3, 2.5, 2.7, and 2.9 ms), and late (mean MEP at ISIs 4.1, 4.3, and 4.5 ms) epochs. The intensity of the first conditioning suprathreshold stimulus (S1) is adjusted to produce a baseline MEP of ~1 mV when given alone and is followed by a second subthreshold stimulus (S2) that was set to 70% of RMT (*Batsikadze et al., 2013*). For each ISI, 15 pulses were recorded. Another 15 pulses were recorded for the control MEPs, in which the suprathreshold stimulus (S1) was given alone and adjusted to achieve a baseline MEP of ~1 mV. The pairs of stimuli were organized in blocks in which each ISI and one test pulse were represented once and were pseudorandomized. This TMS paired-pulse protocol (a first suprathreshold stimulus

and a second subthreshold stimulus) has facilitatory effects on MEP peaks (*Ziemann et al., 1998*) that occur at ISIs of about 1.3, 2.6, and 4.2 ms. This effect is suggested to be produced as a result of elicited I-waves (indirect waves: descending volleys produced by indirect activation of pyramidal tract neurons via presynaptic neurons) by the subthreshold S2 and is controlled by GABA-related neural circuits (*Ziemann et al., 1998*; *Hanajima et al., 2002*; *Paulus et al., 2008*).

### Short-latency afferent inhibition

SAI is a TMS protocol coupled with peripheral nerve stimulation and is based on the concept that peripheral somatosensory inputs have an inhibitory effect on motor cortex excitability at short intervals (e.g. 20–40 ms; *Di Lazzaro et al., 2005b*). SAI has been linked with cholinergic (*Di Lazzaro et al., 2000*) and GABAergic systems (*Di Lazzaro et al., 2005a*) at the cortical level. In this protocol, single-pulse TMS serves as test stimulus and is adjusted to evoke a MEP response with a peak-to-peak amplitude of approximately 1 mV. The conditioning afferent stimuli were single pulses (200 μs) of electrical stimulation applied to the right ulnar nerve at the wrist level (cathode proximal) through bipolar electrodes connected to a Digitimer D185 stimulator (Digitimer Ltd., Welwyn Garden City, UK). The conditioning afferent stimuli were applied with an intensity of ~2.5–3 times of perceptual threshold adjusted to evoke a minimal visible twitch of the thenar muscles (*Di Lazzaro et al., 2000*), followed by a single TMS pulse (test stimulus) applied over the motor cortical representation of the right ADM. The stimuli were applied in blocks containing the test stimulus alone (control condition) and two paired-stimuli blocks with ISIs of 20 and 40 ms in pseudorandomized order. Each block was repeated 20 times, resulting in a total of 60 trials.

### Experimental procedure

Cortical excitability was monitored right after the behavioral/EEG measurements at a same fixed time in both, *sufficient sleep* and *sleep deprivation* sessions. In each session, participants were seated comfortably in a reclining chair, with a pillow resting under the right arm and a vacuum-pillow around the neck to prevent head movement. First, the hotspot (the coil position over the primary motor area that produces the largest MEP in the right ADM with a given medium TMS intensity) was identified with TMS and marked with a water-proof pen. The stimulation intensity was then adjusted to evoke MEPs with a peak-to-peak amplitude of an average of 1 mV. Following this step, RMT and AMT were obtained. A 10 min break was allowed after recording AMT in order to avoid an effect of muscle contraction on the next measurements. After the break, the following TMS protocols were measured to monitor cortical excitability: SAI, SICI-ICF, I-wave facilitation, and I-O curve. The order of measures was randomized except for the I-O curve, which was always the last measure as it required applying high intensities of TMS, which might induce aftereffects on excitability. In the case of single test pulse-generated MEP alterations of >20% during the session in the double pulse conditions, stimulation intensities were adjusted (*Kuo et al., 2017*). Participants were visually monitored to prevent them from closing the eyes due to sleep pressure. Each cortical excitability session took 60–70 min. All TMS protocols were conducted with a PowerMag magnetic stimulator (Mag & More, Munich, Germany) through a figure-of-eight magnetic coil (diameter of one winding, 70 mm; peak magnetic field, 2T), held 45° to the midline and applied over the left primary motor cortex.

## Neuroplasticity

Electrical direct current was applied through a pair of saline-soaked surface sponge electrodes (35 cm$^2$) and delivered through a battery-driven constant current stimulator (neuroConn GmbH, Ilmenau, Germany). The target electrode was fixed over the motor-cortical representation area of the right ADM as identified by TMS, and the reference electrode was placed over the contralateral supraorbital area. The distance on the scalp between the edges of the electrodes was kept at a minimum of 6 cm to reduce shunting of current through the scalp (*Nitsche et al., 2007*). Based on the randomized condition, anodal, cathodal, or sham tDCS with 1 mA intensity was applied for 7 min with 15 s ramp up/down at the beginning and end of stimulation. For the sham condition, stimulation was delivered for 30 s, with a 30 s ramp up and down. Using this procedure, participants are not able to distinguish between real and sham tDCS (*Ambrus et al., 2012*). TMS intensity was set to evoke MEPs of approximately 1 mV peak-to-peak amplitude and single-pulse MEPs were then obtained.

## Experimental procedure

Participants were randomly assigned to the anodal tDCS (N=15) or cathodal tDCS (N=15) groups. Each participant attended four sessions of tDCS (active and sham tDCS after 'sufficient sleep' and 'sleep deprivation') in a counterbalanced order. Order of stimulation was similar across sessions for each participant. tDCS sessions were conducted after the cortical excitability measurement at a fixed starting time and took roughly 90 min in total. In each session, participants were seated comfortably in a reclining chair, with a pillow positioned under the right arm and a vacuum-pillow around the neck to prevent head movement. First, baseline cortical excitability was measured by inducing MEPs over the left M1 representation of the target muscle (right ADM) with a given TMS intensity. The hotspot region was already identified and marked in the cortical excitability part. Stimulation intensity was adjusted to reach a peak-to-peak MEP amplitude of 1 mV (SI1mV), which was then used for the remaining measurements. Following a baseline measurement of 25 MEPs, 7 min of active (anodal or cathodal, depending on group assignment) or sham stimulation was delivered. Right after, MEP measurements were conducted immediately in epochs of every 5 min for up to 30 min after tDCS (seven total epochs). About 5 min after the last MEP measurement (30 min following tDCS), the second tDCS intervention (active or sham) started with the same experimental procedure. Based on the previous works, this tDCS protocol induces polarity-specific short aftereffects up to 30 min after stimulation (**Nitsche and Paulus, 2001**), and thus applying the second intervention after 30 min is feasible. At the end of each session, participants completed a side effect survey to rate the presence and severity of potential adverse effects during stimulation and also asked to guess the stimulation intensity they received (i.e. 0 mA intensity or 1 mA intensity) to evaluate blinding efficacy. The tDCS intervention including both, active and sham stimulation took around 90 min in each session.

## Behavioral measures

### Motor learning

The SRTT was used to measure implicit motor learning in participants. Performance on this task is associated with increased activity and cortical excitability of the motor, premotor, and supplementary motor areas and early learning affects primarily the primary motor cortex (**Honda et al., 1998**; **Nitsche et al., 2003a**; **Schendan et al., 2003**). In brief, the SRTT consisted of eight blocks in which participants should respond to a visually cued stimuli sequence on a computer screen with the respective finger positioned on a keyboard as fast and accurately as possible. Participants are instructed to push the respective button with the respective finger of the right hand (index finger for Button 1, middle finger for Button 2, ring finger for Button 3, and little finger for Button 4). In blocks 1 and 6, the sequence of dots followed a pseudorandom order and in the other blocks, the order of stimuli follows an implicit sequence (e.g. A–B–A–D–B–C–D–A–C–B–D–C). The averaged RT difference in block 5 (sequence order) vs block 6 (random order) is the primary measure of motor *learning acquisition* as it indicates response to sequence learning vs sequence learning-independent performance. The RT difference between block 6 (random order) and block 7 (sequenced order) is suggested to indicate additionally *learning retention*. In addition to RT, which is the major indicator of *implicit* motor learning, RT variability and accuracy were also calculated as outcome variables. Participants were not told about the repeating sequence and at the end of the session, they were asked whether they noticed a sequence and if so, to write the sequence in order to assess explicit learning of the task. In such a case, the data were excluded from the final analysis. Two different sequences of the task, with no overlapping parts, and comparable difficulties, were presented in the two sessions in a counterbalanced order.

### Working memory

A three-back version letter of the task (**Mull and Seyal, 2001**) was used to measure working memory. In this task, participants should indicate whether a letter presented on the screen (the 'target letter') matched the letter previously presented (the 'cue' letter). 'Hits' (correct responses) were defined as any letter identical to the one presented three trials back. Stimuli were pseudorandom sequences of 10 letters (A–J) presented at a fixed central location on a computer screen. Each letter was visible for 30 ms with a 2000 ms ISI, making the difficulty level of the task high. The letters were presented in black on a white background and subtended 2.4 cm (when viewed at 50 cm eye to screen distance).

Participants completed 2 blocks consisting of 44 (practice block) and 143 trials (main block), respectively, resulting in a total number of 187 trials. A short break (5–20 s) between blocks was provided to allow participants to rest. Two different versions of the task were employed in two sessions (sufficient sleep vs sleep deprivation sessions), and condition order was randomized across participants. Accuracy, *d* prime (the proportion of hits rate minus the proportion of false alarm rate) and 'Hits' RT measures were the outcome measures.

## The Stroop color-word task

The Stroop interference task is a neuropsychological test extensively used for measuring selective attention, cognitive inhibition, and information processing speed (*Treisman and Fearnley, 1969*; *Grundey et al., 2015*). We used a computerized Stroop color/word test similar to the Victoria version, based on the previous studies (*Grundey et al., 2015*). This task includes three blocks, the Stroop word, the Stroop color, and the Stroop color-word task. In the Stroop word, the color names were written in black, and in the Stroop color, capital XXXs were presented in red, green, yellow, and blue ink, and participants had to respond with the corresponding keys. In the Stroop color-word task, participants were presented with either 'congruent' or 'incongruent' color words. In the incongruent trials, the color of the ink in which the word was displayed was different from the meaning of the word (e.g. the word 'red' was written in blue) while in the congruent trials both, word and color of the ink, were identical. Stimuli were presented on a screen with black background for 2000 ms with a 500 ms ISI. The size of the stimuli was 1.4 cm at approximately 50 cm eye-to-screen distance. A response box with only four keys, colored in red, blue, yellow, and green, was placed in front of the subjects and they had to press the corresponding key of the color in which the word was written. The Stroop interference block included 40 congruent and 120 incongruent trials, resulting in a total of 160 trials. The reason for the higher number of trials in the Stroop block was to increase the power of the EEG analyses.

## AX-continuous performance test

The AX-CPT is used for assessing attentional functioning (sustained or transient attention), or executive control, depending on the applied versions, which include baseline, proactive control, and reactive control (*Smid et al., 2006*; *Gonthier et al., 2016*). A baseline version of the task was used, which is shorter (around 15 min), less demanding, and measures transient attention (*Smid et al., 2006*). In this task, visual stimuli were white letters on a dark background appearing one at a time on a computer screen for 150 ms each with a 2000 ms ISI. Subjects were instructed to press a button with the right index finger whenever the letter A (correct cue) was followed by the letter X (correct target) as quickly and accurately as possible. All other sequences were to be ignored, including sequences in which an incorrect cue (designated 'B', but comprising all letters other than A or X) was followed by the target letter (X), or sequences in which a correct cue (A) was followed by an incorrect target (designated 'Y', but comprising all letters other than A or X). The AX sequences are presented with a high probability, to guarantee a strong response bias. The tasks consisted of 240 pairs of letters (480 trials) with 40% 'AX', 40% 'BY', 10% 'BX', and 10% of 'AY'. Accuracy and RT were recorded for the target trials.

## Procedure

Participants performed the tasks in two versions randomized across the sufficient sleep and sleep-deprivation sessions with at least 2 weeks intervals. The order of tasks was counterbalanced across participants. All tasks (SRTT, N-back, Stroop, and AX-CPT) were presented on a computer screen (15.6 in. Samsung) via E-prime software (*Schneider et al., 2002*), the viewing distance from the monitor was approximately 50 cm. The tasks were conducted in a soundproof electromagnetic shielded room during EEG recording. The session took about 1 hr and 45 min including cap-preparation time and cleaning the head. Following this part, participants were instructed to remove the gel from their hair and head skin and were guided to the TMS lab for monitoring cortical excitability and neuroplasticity induction.

## Electroencephalogram

### EEG recording

EEG recording included resting-state measurements, which consisted of eyes open and closed states alternating every 2 min for 4 min, and task-based measurements. EEG was recorded from 64 scalp electrodes with 2 additional horizontal and vertical electro-oculogram electrodes (HEOG, VEOG) to measure horizontal and vertical eye movements. The electrodes were positioned according to the international 10–20 system using the NeurOne Tesla EEG amplifier (Bittium, NeurOne, Bittium Corporation, Finland) with a sampling rate of 1000 Hz. The scalp electrodes sites included: Fp1, Fp2, F7, F3, Fz, F4, F8, FC5, FC1, FC2, FC6, T7, C3, Cz, C4, T8, FPz, CP5, CP1, CP2, CP6, PO9, PO5, P7, P3, Pz, P4, P8, FCz, O1, Oz, O2, AF7, AF3, AF4, AF8, F5, F1, F2, F6, TP9, FT7, FC3, FC4, FT8, TP10, C5, IZ, PO10, C6, TP7, CP3, CPz, CP4, TP8, P5, P1, P2, P6, PO7, PO3, POz, PO4, and PO8, and were mounted on the head with a cap (EASYCAP GmbH, Herrsching, Germany). The reference electrode was positioned on FCz, and the ground electrode was placed at the AFz position. The electrodes were connected to the head using high-viscosity electrolyte gel (SuperVisc, Easycap, Herrsching, Germany). All impedances were kept below 10 kΩ throughout the experimental sessions. EEG data were collected in a shielded room, and no spectral peaks at 50 Hz were observed. Raw EEG data were recorded and stored for offline analysis using BrainVision Analyzer 2.1 (Brain Products GmbH, München, Germany).

### EEG data analysis

EEG recordings were band-pass filtered offline between 1 and 30 Hz (48 dB/Octave) and re-referenced to an average reference. The VEOG and HEOG signals were used to correct for eye movement artifacts in ERP recordings using the Gratton and Coles method (*Gratton et al., 1983*) embedded in the BrainVision Analyzer 2.1. EEG data were then time-locked to the stimulus of interest onset in each task. Epochs started 100 ms before stimulus onset and ended 700 ms after stimulus onset in the SRTT, 100 ms before the target onset and ended 1000 ms after target onset in the three-back and AX-CPT tasks, and 100 ms before stimulus onset and ended 1000 ms after stimulus onset in the Stroop task (both congruent and incongruent trials). Epochs were baseline-corrected using a –100 to 0 ms time window. Artifacts were identified using a combination of automated (artifacts greater than 100 µV peak-to-peak) and manual selection processes. Segments were removed based on this automatic selection, and visual inspection to identify artifacts due to sources of non-neurogenic activity. The remaining epochs were averaged for calculating the average ERP. Average ERP of blocks 5, 6, 7 in the SRTT task was based on 120 trials per block. In the N-back and AX-CPT tasks, average ERP of hits (correct response) was based on 40 and 96 trials, respectively. In the Stroop task, the average ERP of congruent and incongruent trials was based on 40 and 120 trials, respectively. For the analyses, the following averaged components were investigated: (1) the P300 at electrodes Pz, Cz, and P3 within a time window of 300–600 ms after stimulus onset in the SRTT learning blocks (block 5, 6, 7), (2) the P300 at electrodes Fz and Cz within a time window of 300–600 ms (*Picton, 1992*; *Kok, 2001*) after target stimulus onset in the three-back task, (3) the N200 and N450 at electrodes Fz and Cz within time windows of 200–300 ms and 400–550 ms, respectively after congruent and incongruent trials onset (*Feroz et al., 2017*) in the Stroop task, and (4) the P300 at electrodes Fz and Cz within a time window of 300–600 ms (*Picton, 1992*; *Tekok-Kilic et al., 2001*) after target onset (when target letter X was preceded by cue A) in the AX-CPT task. The time windows were selected based on previous studies and designated as the maximum positive or negative deflection occurring at the post-stimulus latency window. A fast Fourier transform analysis (Hanning window length: 10%) was performed on the epochs to obtain spectral power levels in the beta (13–30 Hz), alpha (7–13 Hz), theta (4–7 Hz) ,and delta (1–4 Hz) range.

## Statistical analysis

The primary statistical procedures were repeated measure mixed model and/or within-subject ANOVA. Post hoc tests were conducted conditional on significant results of the ANOVAs. The data analyses were conducted independently for each dataset, and corrections for multiple comparisons were performed separately for each analysis.

## Cortical excitability

For the TMS protocols with a double-pulse condition (i.e. SICI-ICF, I-wave facilitation, SAI), the resulting mean values were normalized to the respective single-pulse condition. First, mean values were calculated individually and then inter-individual means were calculated for each condition. For the I-O curves, absolute MEP values were used. To test for statistical significance, repeated measures ANOVAs were performed with ISIs, TMS intensity (in I-O curve only), and condition (sufficient sleep vs sleep deprivation) as within-subject factors and MEP amplitude as the dependent variable. In case of significant results of the ANOVA, *post hoc* comparisons were performed using Bonferroni-corrected t-tests to compare mean MEP amplitudes of each condition against the baseline MEP and to contrast sufficient sleep vs sleep deprivation conditions.

## Neuroplasticity

The mean peak-to-peak amplitude of the 25 MEPs obtained for each timepoint (BL, 0, 5, 10, 15, 20, 25, 30 min after tDCS) was calculated and averaged for active and sham tDCS in the sufficient sleep and sleep deprivation conditions. To determine if individual baseline measures differed within and between sessions, SI1mV and Baseline MEP were entered as dependent variables in a mixed-model ANOVA with session (four levels) and condition (sufficient sleep vs sleep deprivation) as within-subject factors, and group (anodal vs cathodal) as between-subject factor. The mean MEP amplitude for each measurement timepoint was normalized to the session's baseline (individual quotient of the mean from the baseline mean) resulting in values representing either increased (>1.0) or decreased (<1.0) excitability. Individual averages of the normalized MEP from each timepoint were then calculated and entered as dependent variables in a mixed-model ANOVA with repeated measures with stimulation condition (active, sham), timepoint (eight levels), and sleep condition (normal vs deprivation) as within-subject factors and group (anodal vs cathodal) as between-subject factor. In case of significant ANOVA results, *post hoc* comparisons of MEP amplitudes at each time point were performed using Bonferroni-corrected t-tests to examine if active stimulation resulted in a significant difference relative to sham (comparison 1), baseline (comparison 2), the respective stimulation condition at sufficient sleep vs sleep deprivation (comparison 3), and the between-group comparisons at respective timepoints (comparison 4).

## Behavioral task performance

Means of RT, RT variability, and accuracy for SRTT blocks 5, 6, and 7 were calculated. Trials with wrong responses, as well as those with RTs of less than 150 ms (*Collins and Long, 1996*; *Mella et al., 2015*) or more than 3000 ms, and trials that are deviated by three standard deviations or more from the average individual response time, were discarded. The mean RT, RT variability, and accuracy of blocks were entered as dependent variables in repeated measures ANOVAs with block (5 vs 6, 6 vs 7) and condition (sufficient sleep vs sleep deprivation) as within-subject factors. Because the RT differences between blocks 5 vs 6 and 6 vs 7 were those of major interest, *post hoc* comparisons were performed on RT differences between these blocks using paired-sample t-tests (two-tailed, p<0.05) without correction for multiple comparisons. For three-back, Stroop, and AX-CPT tasks, mean and standard deviation of RT and accuracy were calculated and entered as dependent variables in repeated measures ANOVAs with sleep condition (sufficient sleep vs sleep deprivation) as the within-subject factor. For significant ANOVA results, *post hoc* comparisons of dependent variables were performed using paired-sample t-tests (two-tailed, *P*<0.05) without correction for multiple comparisons.

## Correlational analyses

To assess the relationship between induced neuroplasticity and motor sequence learning, and the relationship between cortical excitability and cognitive task performance we used bivariate linear regression analysis (Pearson's correlation, two-tailed). For the first correlation, we used individual grand-averaged MEP amplitudes obtained from anodal and cathodal tDCS pooled for the timepoints between 0 and 20 min after interventions, and individual motor learning performance (i.e. BL6-5 and BL6-7 RT difference) across sleep conditions. For the second correlation, we used individual grand-averaged MEP amplitudes obtained from each TMS protocol and individual accuracy/RT obtained from each task across sleep conditions. No correction for multiple comparisons was done for correlational analyses as these were secondary exploratory analyses.

## Electroencephalogram

EEG data preprocessing and analysis were described in the previous section. For the resting-state data, brain oscillations at mid-central electrodes (Fz, Cz, F3, F4) were analyzed with a 4×2 ANOVA with location (Fz, Cz, F3, F4) and sleep condition (sufficient sleep vs sleep deprivation) as the within-subject factors. For all tasks, individual ERP means were grand-averaged and entered as dependent variables in repeated measures ANOVAs with sleep condition (sufficient sleep vs sleep deprivation) as the within-subject factor. *Post hoc* comparisons of grand-averaged amplitudes were performed using paired-sample t-tests (two-tailed, $P<0.05$) without correction for multiple comparisons.

## Acknowledgements

We appreciate Nicole Rück, Ensiyeh Ghasemian-Shirvan, Elham Ghanavati, and Mohsen Mosaybi-Samani for the stay overnight during the course of the experiment. We also thank Tobias Klimek and Tobias Blanke for their technical support. MAN is supported by a grant from the German Ministry of Research and Education (GCBS grant 01EE1403C).

## Additional information

### Competing interests

Michael A Nitsche: is a member of the Scientific Advisory Boards of Neuroelectrics and NeuroDevice. The other authors declare that no competing interests exist.

### Funding

No external funding was received for this work.

### Author contributions

Mohammad Ali Salehinejad, Conceptualization, Data curation, Formal analysis, Investigation, Methodology, Software, Validation, Visualization, Writing – original draft; Elham Ghanavati, Data curation, Visualization, Writing – review and editing; Jörg Reinders, Jan G Hengstler, Formal analysis, Writing – review and editing; Min-Fang Kuo, Conceptualization, Supervision, Writing – review and editing; Michael A Nitsche, Conceptualization, Methodology, Project administration, Resources, Supervision, Writing – review and editing

### Author ORCIDs

Mohammad Ali Salehinejad http://orcid.org/0000-0003-1913-4677
Elham Ghanavati http://orcid.org/0000-0001-5944-8123
Min-Fang Kuo http://orcid.org/0000-0001-9955-0237
Michael A Nitsche http://orcid.org/0000-0002-2207-5965

### Ethics

Human subjects: This study conformed to the Declaration of Helsinki guidelines and was approved by the Institutional Review Board (ethics code: 99). Participants gave informed consent and received monetary compensation.

### Decision letter and Author response

Decision letter https://doi.org/10.7554/eLife.69308.sa1
Author response https://doi.org/10.7554/eLife.69308.sa2

## Additional files

### Supplementary files

• Supplementary file 1. *Note*: MEP: motor-evoked potentials; $SI_{1mv}$ (%): maximum stimulator output (%MSO) required for the SI1mV MEP amplitude; I-O curve: input-output curve; SICI-ICF: short-latency intracortical inhibition and facilitation; SAI: short-latency afferent inhibition; RMT: resting motor threshold; AMT: active motor threshold.

• Supplementary file 2. *Note*: MEP: motor-evoked potentials; sleep condition: =sufficient sleep and sleep deprivation; $SI_{1mv}$ (%): maximum stimulator output (%MSO) required for the SI1mV MEP amplitude; I-O curve: input-output curve; SICI-ICF: short-latency intracortical inhibition and facilitation; SAI: short-latency afferent inhibition; RMT: =resting motor threshold; AMT: active motor threshold.

• Supplementary file 3. *Note*: tDCS: transcranial direct current stimulation; MEP:motor-evoked potentials; $SI_{1mv}$ (%): maximum stimulator output (%MSO) required for the SI1mV MEP amplitude.

• Supplementary file 4. The presence and intensity of the side-effects were rated on a numerical scale ranging from 0 to 5, 0 representing no and 5 extremely strong sensations. Data are presented as mean ± SD.

• Supplementary file 5. The presence and intensity of reported side-effects during tDCS were analyzed by repeated measures mixed-model ANOVAs with sleep condition (sufficient sleep vs sleep deprivation) and tDCS state (active vs sham) as the within-subject factors and group (anodal, cathodal) as the between-subject factor. Pairwise comparisons are calculated using Student's t-test. n=30 (15 per group).

• Transparent reporting form

### Data availability

The data files generated in this study are publicly available at https://osf.io/kve6d via Open Science Foundation (OSF).

The following dataset was generated:

| Author(s) | Year | Dataset title | Dataset URL | Database and Identifier |
|---|---|---|---|---|
| Salehinejad MA | 2021 | Sleep, brain physiology and cognition | https://doi.org/10.17605/OSF.IO/KVE6D | Open Science Framework, 10.17605/OSF.IO/KVE6D |

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

# Appendix 1

## Supplementary material

### 1. Cortical excitability supplementary results

#### 1.1. TMS protocols threshold values

Baseline MEP values of single-pulse conditions and other control conditions, as well as subject-specific baseline sensitivity to TMS (SI1mV), are summarized in *Supplementary file 1A*. For each TMS protocol, single-pulse condition MEPs (control condition) and the SI1mV obtained at each sleep condition were compared by a repeated measures ANOVA with sleep condition (sufficient sleep, sleep deprivation) as the within-subject factor. The results of the respective ANOVAs show that the mean values of the single pulse-elicited MEP, %MSO for RMT and AMT, and $SI_{1mv}$ did not significantly differ between the sleep conditions (*Supplementary file 1B*). A trendwise difference was found between $SI_{1mv}$ values of baseline MEP, with lower values after sleep deprivation compared to sufficient sleep ($P=0.054$). Furthermore, for the I-O curve protocol, a significant difference between MEP amplitudes at RMT intensity ($t=2.42$, $P=0.022$) was revealed.

### 2.Neuroplasticity supplementary results

#### 2.1.Baseline MEP difference

The average baseline MEPs (absolute value) as well as the TMS stimulus intensity required for 1 mV amplitude (SI1mV) obtained for each tDCS condition (anodal, cathodal, sham) for each group are summarized in *Supplementary file 1C*. The baseline MEP values obtained from each sleep condition (sufficient sleep, sleep deprivation) for both groups (anodal, cathodal) from each tDCS state (active, sham) were analyzed with a 2×2×2 factorial ANOVA. The results show no interaction of sleep condition × group × tDCS state ($F_1=1.48$, p=0.234), or sleep condition × group ($F_1=0.49$, p=0.486), or sleep condition × tDCS state ($F_1=0.65$, p=0.426), or group × tDCS state ($F_1=0.36$, p=0.550). The main effects of group ($F_1=0.05$, p=0.819), sleep condition ($F_1=0.06$, p=0.811), and tDCS state ($F_1=0.17$, p=0.680) were not significant neither. The same 2×2×2 factorial ANOVA was conducted for SI1mV values, and the results show no interaction of sleep condition × group × tDCS state ($F_1=0.25$, p=0.616), sleep condition × group ($F_1=1.06$, p=0.311), sleep condition × tDCS state ($F_1=0.45$, p=0.505), and main effect of group ($F_1=0.88$, p=0.355). The interaction of group × tDCS state ($F_1=4.73$, p=0.038), and the main effects of sleep condition ($F_1=6.65$, p=0.015), and tDCS state ($F_1=5.48$, p=0.026) were, however, significant. Pairwise comparisons of stimulation state (active vs sham) in the sleep conditions with post hoc t-tests revealed significant differences of SI1mV amplitudes between sufficient sleep and sleep deprivation conditions for both, active tDCS ($t=2.58$, p=0.015) and sham tDCS ($t=2.46$, p=0.020). When stimulation states were compared at the same sleep condition (active vs sham, sufficient sleep; active vs sham, sleep deprivation), a marginal significant difference in SI1mV amplitudes was observed between active and sham tDCS in the sufficient sleep ($t=2.06$, p=0.048), but not sleep deprivation ($t=1.17$, p=0.252) conditions.

#### 2.2. Reported tDCS side effects

The reported side effects during each tDCS session (average ± SD) after sufficient sleep and sleep deprivation are summarized in *Supplementary file 1D*. The results of the 2 (group: anodal, cathodal) × 2 (sleep condition) 2 ✖ (tDCS state: active, sham) factorial ANOVA conducted for each side effect showed no interaction or main effects, except for a significant main effect of tDCS state for itching, and tingling (*Supplementary file 1E*). Pairwise comparisons of itching and tingling ratings with post hoc t-tests revealed a significantly higher rating for itching sensation during anodal tDCS compared to the sham condition only after sleep deprivation ($t=2.41$, p=0.030). When the ratings were compared regardless of stimulation polarity (i.e. active tDCS vs sham tDCS), a significantly higher rating of the itching sensation was observed between active and sham tDCS after both sufficient sleep ($t=2.85$, p=0.008) and sleep deprivation ($t=3.06$, p=0.005). The intensity of the reported side effects was in general low.

#### 2.3.tDCS blinding efficacy

To explore blinding efficacy we asked participants to guess whether they received real tDCS (1 mA) or sham tDCS (0 mA) after each stimulation condition across sleep conditions. Using the Chi-square test for associations, we explored whether participants in each group (anodal, cathodal) could correctly discern each real stimulation condition from its respective sham condition in the sufficient sleep and sleep deprivation sessions. The results of the respective Chi-square tests show no significant

differences of participants' guesses between each real stimulation and sham stimulation in both, anodal ($\chi^2$=0.682, p=0.409; $\chi^2$=0.085, p=0.770) and cathodal groups ($\chi^2$=0.00, p=1.000; $\chi^2$=0.268, p=0.605), and the whole group ($\chi^2$=0.557, p=0.448; $\chi^2$=0.007, p=0.993) (All).

## 3. Implicit motor learning

### 3.1 Absolute reaction time

We also analyzed SRTT task performance based on the absolute RT values. The results of the 2×3 ANOVA showed a significant interaction of sleep condition ✕ block ($F_{1.89}$=3.43, p=0.042, $\eta p^2$=0.11), and significant main effects of sleep condition ($F_1$=51.95, p<0.001, $\eta p^2$=0.64), and block ($F_{1.94}$=30.17, p<0.001, $\eta p^2$=0.51). Post hoc comparisons of blocks revealed a significantly faster RT at blocks 5 and 7 and longer RT at block 6 after sufficient sleep, but not sleep deprivation (*Figure 5—figure supplement 1a*). Baseline block and block 6 RT (two values), which contain stimuli in random order were also compared, and the results of the 2×2 ANOVA showed a significant main effect of sleep condition ($F_1$=2.27, p<0.001), baseline block ($F_1$=10.16, p=0.003) and their interaction ($F_1$=12.72, p=0.002). These results show a generally slower RT after sleep deprivation, compared to sufficient sleep (*Figure 5—figure supplement 1a*).

### 3.2. Error rate

The number of errors in the learning blocks in the respective sleep conditions was analyzed as well. The results of the 2×3 ANOVA showed a significant interaction of sleep condition × block ($F_{1.96}$=8.49, p=0.001, $\eta p^2$=0.23) and a significant main effect of sleep condition ($F_1$=6.49, p=0.016, $\eta p^2$=0.18), but not block ($F_{1.73}$=1.02, p=0.365). Post hoc t-tests showed a significantly higher number of committed errors at block 6 compared to block 5 only after sleep deprivation (*Figure 5—figure supplement 1b*). Furthermore, when every single block was compared across sleep conditions, the number of committed errors was significantly higher at BL 4, 6, 7, and 8 after sleep deprivation (*Figure 5—figure supplement 1c*).

### 3.3. RT variability

The results of the 2×3 ANOVA showed a significant interaction of sleep condition ✕ block ($F_{1.55}$=4.57, p=0.023, $\eta p^2$=0.13) and main effects of sleep condition ($F_1$=16.72, p<0.001, $\eta p^2$=0.36) and block ($F_{1.81}$ = 4.64, p=0.016) on RT variability. Post hoc t-tests showed no significant difference between RT variation of at block 6 compared to block 5 across sleep conditions. The RT variability from BL 6–7 was, however, significantly higher after sleep deprivation. Furthermore, when every single block was compared across sleep conditions, RT variability was significantly higher at each block (BL 5, 6, 7) after sleep deprivation compared to sufficient sleep (*Figure 5—figure supplement 1d*).

### 3.4. EEG supplementary results

Further analyses were conducted for electrodes of centroparietal regions close to the electrodes of interest (C1, C2, P1, P2). These revealed larger P300 amplitudes at block 6 vs block 5 and 7 after sufficient sleep. For the C1 electrode, the results of the respective ANOVA showed a significant main effect of sleep condition ($F_1$=9.54, p=0.001, $\eta p^2$=0.25) and block ($F_{1.48}$=3.49, p=0.050, $\eta p^2$=0.11), but no interaction of these factors on P300 amplitudes. Post hoc comparisons showed a significantly larger P300 amplitude in all learning blocks, including block 6, after sufficient sleep compared to sleep deprivation ($t_{BL5}$=2.10, p=0.044, $t_{BL6}$=3.23, p=0.003, $t_{BL7}$=2.48, p=0.019; *Figure 5—figure supplement 2*). No significant difference between BL 6–5 and BL 6–7 was observed in either condition. For the electrode C2, the results of the ANOVA showed a significant interaction of sleep condition ✕ block ($F_2$=3.32, p=0.043; $\eta p^2$=0.10), and main effects of sleep condition ($F_1$=5.02, p=0.033, $\eta p^2$=0.15) and block ($F_{1.42}$=4.76, p=0.023, $\eta p^2$=0.14). Post hoc comparisons of learning block showed a significantly higher P300 amplitude at block 6 compared to blocks 5 and 7 only after sufficient sleep ($t_{6-5}$=2.74, p=0.010, $t_{6-7}$=2.64, p=0.013) but not sleep deprivation ($t_{6-5}$=0.70, p=0.485, $t_{6-7}$=1.92, p=0.064). The P300 amplitudes were also significantly larger at blocks 6 and 7 after sufficient sleep compared to the sleep deprivation ($t_{BL6}$=2.23, p=0.034, $t_{BL7}$=2.21, p=0.035; *Figure 5—figure supplement 2*). Finally, the results of the ANOVA conducted for the electrode P1 showed a significant main effect of learning blocks ($F_{1.40}$=5.68, p=0.013, $\eta p^2$=0.16) but not sleep condition ($F_1$=1.54, p=0.22) or interaction of sleep condition × block ($F_{1.15}$=2.12, p=0.152) on the P300 amplitude. Post hoc comparisons of P300 amplitudes within and between conditions showed a significantly larger component at block 6 vs 5 (t=4.21, p<0.001) and 6 vs 7 (t=4.71, p<0.001) only after sufficient sleep. Similarly, for electrode P2, a significant main effect of learning blocks ($F_{1.39}$=7.79, p=0.004, $\eta p^2$=0.21), but not sleep condition ($F_1$=1.78 = 1, p=0.21), or interaction of sleep

condition × block ($F_{1,20}$=2.73, p=0.073) were found for the P300 amplitude. Post hoc comparisons of the P300 amplitude within and between conditions showed that the P300 amplitude was significantly larger at block 6 vs 5 (t=4.27, p<0.001) and 6 vs 7 (t=5.17, p<0.001) only after sufficient sleep (*Figure 5—figure supplement 2*).

## 4. Working memory and attention tasks

For working memory performance, we also calculated variability of RT at a secondary outcome measure. The result of the within-subject design ANOVA revealed a significant main effect of sleep conditions RT variability of hits ($F_1$=4.78, p=0.037). Post hoc Student's t-tests showed a significantly enhanced WM performance with significantly more RT variability after sufficient sleep, which could be due to an accuracy-RT trade-off. In the Stroop task, we investigated performance accuracy and ERP components at electrode Cz as well. The results of respective ANOVAs showed a significant main effect of sleep condition on the overall accuracy of the Stroop stage ($F_1$=6.32, p=0.018; $\eta p^2$=0.18), accuracy of congruent trials ($F_1$=4.77, p=0.037; $\eta p^2$=0.14), and accuracy of incongruent trials ($F_1$=5.03, p=0.029; $\eta p^2$=0.16). Post hoc comparisons of accuracy rate revealed that participants had a significantly higher number of accurate responses to trials in the Stroop stage as well as incongruent and congruent trials (*Figure 5—figure supplement 3a*). For the electrode Cz, the results of the 2 (congruency) × 2 (sleep condition) ANOVA showed only a significant main effect of sleep condition on the N200 ($F_1$=9.03, p=0.006; $\eta p^2$=0.25) but not N450 component (*Figure 5—figure supplement 3b*). Similarly, post hoc Student's t-tests indicated a significantly smaller N200 amplitude, for the incongruent trials only, for the Cz electrode after sleep deprivation as compared to sufficient sleep (*Figure 5—figure supplement 3b*). Finally for the AX-CPT task, we also analyzed ERP components at other potentially relevant electrodes (F3, F4, C3, C4), and a comparable main effect of sleep condition was found on P300 amplitude for electrodes F3 ($F_1$=4.77, p=0.038; $\eta p^2$=0.15), F4 ($F_1$=7.82, p=0.011; $\eta p^2$=0.21), C3 ($F_1$=24.31, p<0.001; $\eta p^2$=0.48), and C4 ($F_1$=7.60, p=0.011; $\eta p^2$=0.22). Post hoc Student's t-tests indicated that sleep deprivation was related to a significantly smaller P300 amplitude in the F3, F4, C3, and C4 electrodes (*Figure 5—figure supplement 3c*).

## 5. Correlational analyses

### 5.1. Correlation between sequence learning and plasticity induction

To explore the association between motor learning and plasticity, we calculated the correlation between the respective parameters (Pearson's correlation, two-tailed). We found a significant negative correlation between enhanced anodal LTP-like plasticity after sufficient sleep and enhanced motor learning (indicated by reduced RT at learning blocks). Specifically, MEP amplitude enhancement after anodal tDCS was negatively correlated with both sequence learning acquisition (blocks 6–5 RT difference; r=−0.558, p=0.031) and sequence learning retention (blocks 6–7 RT difference; r=−0.734, p=0.002). This indicated that LTP-like plasticity effects after sufficient sleep were associated with better sequence learning. No correlation was found between cathodal LTD-like plasticity and sequence learning.

### 5.2. Correlation between cortical excitability, working memory, and attention

To explore the association between physiological parameters of cortical excitability, and cognitive performance, we correlated performance in the three-back letter task, Stroop test and AX-CPT with the respective cortical excitability results. In the three-back letter task, enhanced *d* prime index (a measure of performance accuracy) was positively correlated with cortical facilitation measured by ICF at ISI of 15 ms (i.e. larger MEP at ICF) after having sufficient sleep (r=0.425, p=0.019). Conversely, lower accurate response during sleep deprivation was negatively correlated with converted intracortical inhibition to facilitation (i.e. larger MEP amplitude) measured by SAI at ISI of 40 ms (r=−0.386, p=0.035). This indicates that upscaled cortical facilitation was associated with poor working memory performance.

No correlation was observed between Stroop task outcome measures and cortical excitability measures. For AX-CPT task performance, there was only a significant *negative* correlation between enhanced performance accuracy after sufficient sleep and reduced intracortical inhibition (measured by averaged MEPs of SICI) at the same time (r=−0.372, p=0.043). This indicates that improved task performance (i.e. higher accuracy) were associated with decreased intracortical inhibition after having sufficient sleep. In the sleep deprivation condition, lower performance accuracy and negatively correlated with higher corticospinal excitability (i.e. enhanced MEP at 150% of RMT intensity; r=−0.429, p=0.018).

