## [Editor Report]

This paper provides a comprehensive investigation into the neural effects of sleep deprivation in humans across a broad range of methods and, using non–invasive brain stimulation as well as electrophysiological markers and behavioral measures, the study demonstrates that sleep deprivation results in higher cortical excitability, which may explain the negative impact of sleep deprivation on cognitive processes.

---

## [Decision Letter]

**Decision letter after peer review:**

Thank you for submitting your article "Sleep–dependent upscaled excitability and saturated neuroplasticity in the human brain: From brain physiology to cognition" for consideration by *eLife*. Your article has been reviewed by 2 peer reviewers, and the evaluation has been overseen by a Reviewing Editor and Floris de Lange as the Senior Editor. The reviewers have opted to remain anonymous.

Essential revisions:

1) As mentioned by the reviewers, the power of a comprehensive study lies in the comparison across and integration of different measures/sub–studies. Please integrate the different findings more explicitly, and explain the synergistic value of the different sub–studies. On a related note, please present the design and main aims/hypotheses of the study already early in the manuscript, to aid readers in their understanding of the following sections.

2) Please pay additional attention to the analysis and presentation of the EEG theta activity findings, as mentioned by both reviewers. Besides some caution in presentation/interpretation in terms of synaptic strength, please consider options to differentiate periodic vs. aperiodic activity.

3) Please consider correction for multiple comparisons across the multitude of analyses, and/or emphasize/discuss this point explicitly.

4) Given the multitude of data which has not been analyzed exhaustively, it would be a very beneficial service to the field if the data would be made openly accessible in its entirety.

*Reviewer #1 (Recommendations for the authors):*

– It is great that authors aim to also show the distributional properties of their data by showing single subjects data points in their plots. The way things are plotted, e.g. in Figure 2 make it hard to make out the paired relationships, and also data points are over–plotted. Maybe the size of the markers or their opacity can be adjusted, and the paired relationship be shown. Possibly a plot like R11 here: https://wellcomeopenresearch.org/articles/4–63/v2 could help unify the duplication of right and left columns.

– Figure 5 right column, lines are very hard to see somehow

*Reviewer #2 (Recommendations for the authors):*

Abstract:

– The authors should carefully, and throughout the entire manuscript, avoid using terms of physiological brain processes when referring to their indirect assessments. For instance, EEG theta activity might be considered as an "index of synaptic strength", but the expression "increased synaptic strength measured by EEG theta activity" is misleading.

– The first sentence is a bit difficult to understand: do the authors mean that sleep affects plasticity as a critical mechanism for cognitive functioning?

– The last sentence of the abstract is trivial, and should be refined emphasizing the novel contribution of the work.

– Moreover, I would prefer to have some basic information about the used methods in the abstract.

Introduction:

– The first citations are outdated (2010) and should be replaced by relevant recent reviews.

– Walker and Stickgold, two human researchers, are not the best citation for sleep and cognition in animals (even if the review might include some animal work).

– Together, this start suggests that references should be carefully reassessed and potentially adapted throughout the entire manuscript.

– The authors suggest that the second paragraph of the introduction refers to animal research. However, Kuhn et al. is a study in humans. In general, please indicate more precisely the species, for instance Kuhn et al. humans, de Vivo et al. as far as I remember mice, etc.

– While I understand that the *eLife* format places the methods section at the end, I would still prefer to have some fundamental information about the study design (parallel group? Repeated measures? Number of participants) when reading the beginning of the Results section.

– In general, I miss a bit the clarification of hypotheses; I assume that the authors had hypotheses, for instance on indices of increased cortical excitability and net synaptic strength after sleep deprivation; the authors mostly just state that they monitored or assessed something, without providing a clear framework and directed hypotheses (whereas it is indirectly clear that they had directed hypotheses). Or more generally spoken, the authors tested a quite well refined model of cortical excitability/net synaptic strength and inducibility of LTP– and LTD–like plasticity after sleep deprivation and the relation to parameters of cognition. But rather than summarizing this model in a structured and systematic way, they list numerous single aspects and miss a bit the systematic and clearer picture in the introduction.

Results:

– Second paragraph, not sure whether Bonferroni is appropriate and not too conservative for post–hoc comparisons after a significant ANOVA.

– The significant ANOVA effect might be visualized in Figure 2a.

– The issue of multiple testing is even more relevant for the numerous analyses for numerous different outcomes (ANOVAs) reported in the manuscript. Strictly spoken, there should be a correction for multiple testing on the level of the many outcome parameters, if there was not one a priori primary analysis? Or a MANOVA or something else (I'm not a statistician). Anyhow, the results are convincing, but the approach should be transparent.

– I assume I will understand the design after reading the method section. However, on page 10 in the Results section, I still do not understand if there are parallel gropus or repeated measures or a combination of it.

– It appears to me that numerous correlation analyses do not represent the best way of further exploring the relationships between parameters. Other approached might be considered, such a linear regression models or others (I'm not a statistician, and linear regression might also be inappropriate).

Discussion:

– I would expect that the beginning of the introduction more clearly relates to the tested model/hypotheses, that is provides the context, and not only lists findings (e.g., our findings provide further support for the synaptic homeostasis hypothesis…; or, in line with our initial hypothesis…); moreover, I would expect that the beginning of the introduction more clearly provides information about what represents a replication of prior work, an extension or a novel contribution to the literature.

– There is some redundancy in the discussion, for instance the repetition of the introduction at the beginning of the third paragraph (Previous studies in animals…) that could be deleted.

– On the other hand, it appears that the study provides first findings on LTD–like plasticity in humans – a novelty aspect that could be reemphasized in the third paragraph.

– The finding of a conversion from LTD– to LTP–like plasticity after cathodal stimulation (after sleep deprivation) and the respective potential physiological explanation is of great interest.

– The authors should add that they cannot disentangle the contribution of different sleep stages or fine graded sleep processes, such as sleep slow waves or spindles.

– The authors might add a paragraph of the potential relevance of their findings for healthy performance (e.g. high demanding jobs), aging, and disorders, such as major depression (e.g., synaptic plasticity model, Wolf et al. SMR) or cognitive decline in relation sleep/disrupted sleep.

– Also, the idea of modulating sleep is currently of interest.

– I think the conclusion paragraph is too unspecific; rather specify, e.g. provide further support for… complement… provides first evidence for …

Methods:

– Participants: Please provide information about potential substance use (including smoking).

– Study design: It's the first time (on page 22) that I start to understand the design; please ensure including some fundamental information at an earlier section in the manuscript (such as end of the introduction, or beginning of the Results section).

– Sleep conditions: did participants in the sleep condition slept in their home environment?

[Editors' note: further revisions were suggested prior to acceptance, as described below.]

Thank you for resubmitting your work entitled "Sleep–dependent upscaled excitability, saturated neuroplasticity, and modulated cognition in the human brain" for further consideration by *eLife*. Your revised article has been evaluated by Floris de Lange (Senior Editor) and a Reviewing Editor.

The manuscript has been improved but there are some remaining issues that need to be addressed. While a comprehensive comparison and data integration across the different measures will likely not be possible with the current sample size, and the study can thus be viewed as a parallel investigation of different research questions within the same sample, please revisit the other core points of the previous editorial letter:

Theta: the association between theta and synaptic strength remains unclear. The manuscript now makes this link less explicitly, however, would profit from a more dedicated attempt to clarify potential differences in the human vs. animal literature; the topographic origin of the observed theta findings (previous presentation suggested a stronger occipital rather than midfrontal effects as might be expected from the human literature); and the potential oscillatory nature of the findings (periodic vs. aperiodic).

Multiple comparisons correction: please state more explicitly how correction for multiple comparisons was performed, i.e. within or across analyses.

Data availability: data on the corresponding OSF site does not seem to include raw EEG data. In particular given the still lacking differentiation of periodic vs. aperiodic activity, making this data available would allow others to look into this issue themselves.

---

## [Author Response]

Essential revisions:1) As mentioned by the reviewers, the power of a comprehensive study lies in the comparison across and integration of different measures/sub–studies. Please integrate the different findings more explicitly, and explain the synergistic value of the different sub–studies. On a related note, please present the design and main aims/hypotheses of the study already early in the manuscript, to aid readers in their understanding of the following sections.

Thank you for highlighting this point. We revised the manuscript accordingly. Please see the responses to the reviewers' comments for details. The design, course of study, and main aims/hypotheses of the work are also added to the introduction and we discussed all results more explicitly, please see page 5.

Revised text in the introduction, page 4-6:

“The number of available studies about the impact of sleep deprivation on human brain physiology relevant for cognitive processes is limited, and knowledge is incomplete. […] For the neuroplasticity measures, half of the participants received anodal and the other half cathodal stimulation in a randomized, sham-controlled parallel-group design. Figure 1 shows the detailed course of study.”

2) Please pay additional attention to the analysis and presentation of the EEG theta activity findings, as mentioned by both reviewers. Besides some caution in presentation/interpretation in terms of synaptic strength, please consider options to differentiate periodic vs. aperiodic activity.

Thank you for highlighting this point. We re-analyzed EEG theta activity by adding two more frontal channels and specified that theta activity is specifically increased in the mid-frontocentral area, as suggested by the 1^st^ reviewer, and removed unclear images from figure 4. Please see the revised Figure 4. We also toned down statements about the association between theta activity, and synaptic strength throughout the text. Regarding aperiodic EEG activity, for the present study, our main aim was to obtain EEG data about sleep pressure, which refers to periodic activity. For obtaining information about functions associated with aperiodic activity, this was not in the focus of our interest, but if reviewers insist, we would be willing to add respective analyses. That said, we mentioned in the discussion that our EEG analysis was based on periodic activity and pointed to potential relevance of aperiodic components for future studies.

Revised text 23:

“Third, the power spectral analysis was analyzed with a “band ratio” approach in which each frequency band is analyzed based on periodic, or oscillatory, activity. However, the EEG signal reflects at least some aperiodic components (e.g., power spectrum density) (Donoghue et al., 2020) that were not taken into account, as the focus of our study was on sleep pressure which refers to periodic activity. Considering aperiodic EEG signals could however be relevant for physiological interpretations of sleep-related data in future studies.”

3) Please consider correction for multiple comparisons across the multitude of analyses, and/or emphasize/discuss this point explicitly.

Thank you for highlighting this point. We did corrections for multiple comparisons across all analyses (details are in the methods section). We clarify this on pages 32-34. Please see also our reply to comment 3 of the first reviewer.

4) Given the multitude of data which has not been analyzed exhaustively, it would be a very beneficial service to the field if the data would be made openly accessible in its entirety.

Thank you for highlighting this point. The data is available via the open-access platform of the Open Science Foundation at https://osf.io/kve6d

Reviewer #1 (Recommendations for the authors):– It is great that authors aim to also show the distributional properties of their data by showing single subjects data points in their plots. The way things are plotted, e.g. in Figure 2 make it hard to make out the paired relationships, and also data points are over–plotted. Maybe the size of the markers or their opacity can be adjusted, and the paired relationship be shown. Possibly a plot like R11 here: https://wellcomeopenresearch.org/articles/4–63/v2 could help unify the duplication of right and left columns.

Thank you for your suggestion. We revised Figure 2 as suggested.

– Figure 5 right column, lines are very hard to see somehow.

Thank you. We added the right column to the new figure (Figure 6).

Reviewer #2 (Recommendations for the authors):Abstract:– The authors should carefully, and throughout the entire manuscript, avoid using terms of physiological brain processes when referring to their indirect assessments. For instance, EEG theta activity might be considered as an "index of synaptic strength", but the expression "increased synaptic strength measured by EEG theta activity" is misleading.– The first sentence is a bit difficult to understand: do the authors mean that sleep affects plasticity as a critical mechanism for cognitive functioning?– The last sentence of the abstract is trivial, and should be refined emphasizing the novel contribution of the work.– Moreover, I would prefer to have some basic information about the used methods in the abstract.

Thank you for the points. We revised the abstract as suggested.

– We took this recommendation into account. The respective sentence is revised.

– In the first sentence, we mean the relevance of brain plasticity for learning and memory. We rephrased it.

– We modified the last sentence.

– As far as the abstract word limit (150) allows, we also added some information about the applied methods.

Revised abstract:

“Sleep strongly affects synaptic strength, making it critical for cognition, especially learning, and memory formation. […] Our data suggest that upscaled brain excitability, and altered plasticity, due to sleep deprivation, are associated with impaired cognitive performance.”

Introduction:– The first citations are outdated (2010) and should be replaced by relevant recent reviews.– Walker and Stickgold, two human researchers, are not the best citation for sleep and cognition in animals (even if the review might include some animal work).– Together, this start suggests that references should be carefully reassessed and potentially adapted throughout the entire manuscript.

Thank you. Respective references are updated here and also in the whole manuscript. Please see below:

Revised text, page 3:

“Over the past decade, a strong link has been established between sleep and cognition (Yaffe et al., 2014; Lowe et al., 2017). Adequate sleep is critical for optimal cognitive functions across the lifespan (Carskadon, 2011; Lo et al., 2016) and findings from experimental settings support this critical role of sleep for cognition in animals (Rasch and Born, 2013; Boyce et al., 2017), and humans (Krause et al., 2017) especially for memory consolidation and sequence learning (Stickgold, 2005; Chouhan et al., 2021). As a ubiquitous *physiological* phenomenon, sleep has extensive impacts on brain physiology and especially on parameters relevant for cognition such as brain excitability and plasticity.”

– The authors suggest that the second paragraph of the introduction refers to animal research. However, Kuhn et al. is a study in humans. In general, please indicate more precisely the species, for instance Kuhn et al. humans, de Vivo et al. as far as I remember mice, etc.

Thank you. We removed Kuhn et al. study from this paragraph and modified the references.

Revised text, page 3:

“Previous experimental studies, mostly in nonhuman animals, have linked sleep with synaptic homeostasis. […] This demonstrates that sleep is required for preparing the brain for proper cognitive, motor, and physiological functioning, however, the effect of sleep on specific parameters of *human* brain physiology and their association with cognition and behaviour remains to be further determined.”

– While I understand that the eLife format places the methods section at the end, I would still prefer to have some fundamental information about the study design (parallel group? Repeated measures? Number of participants) when reading the beginning of the Results section.– In general, I miss a bit the clarification of hypotheses; I assume that the authors had hypotheses, for instance on indices of increased cortical excitability and net synaptic strength after sleep deprivation; the authors mostly just state that they monitored or assessed something, without providing a clear framework and directed hypotheses (whereas it is indirectly clear that they had directed hypotheses). Or more generally spoken, the authors tested a quite well refined model of cortical excitability/net synaptic strength and inducibility of LTP– and LTD–like plasticity after sleep deprivation and the relation to parameters of cognition. But rather than summarizing this model in a structured and systematic way, they list numerous single aspects and miss a bit the systematic and clearer picture in the introduction.

Thank you for the suggestions. We revised the introduction to include basic information of the study design/methods as well as the hypotheses.

Revised text, page 4-6:

“In the sleep deprivation paradigm applied in the present study, participants are kept in an extended wakefulness condition for a certain amount of time. […] For the neuroplasticity measures, half of the participants received anodal and the other half cathodal stimulation in a randomized, sham-controlled parallel-group design. Figure 1 shows the detailed course of study.”

Results:– Second paragraph, not sure whether Bonferroni is appropriate and not too conservative for post–hoc comparisons after a significant ANOVA.

Thank you for the comment. We aimed to be consistent regarding the use of post-hoc tests. Nevertheless, we used other tests and the results were non-significant as well (e.g. Sidak, Holm-Sidak). Only using the Fisher’s LSD test showed a significantly higher corticospinal excitability at 150% of RMT intensity after sleep deprivation vs sufficient sleep. We explained in the figure legend that this data shows a trend-wise increase of corticospinal excitability.

Revised text, p.9:

“Figure 2. Corticospinal and corticocortical excitability after sufficient sleep vs sleep deprivation. a, There is a trend of higher corticospinal excitability after the sleep deprivation session compared to sufficient sleep especially at 150% of RMT intensity. The red asterisk refers to significant effects of sleep condition (*p*=0.034) and TMS intensity (*p*<0.001).”

Revised text, p.19:

“In line with our hypothesis and recent previous works, sleep deprivation upscaled parameters of cortical excitability. A trend-wise upscaling was moreover found for corticospinal excitability.”

– The significant ANOVA effect might be visualized in Figure 2a.

We now visualize the overall significant ANOVA effect on Figure 2A and clarified this in the legend. Please see below and the revised Figure 2.

– The issue of multiple testing is even more relevant for the numerous analyses for numerous different outcomes (ANOVAs) reported in the manuscript. Strictly spoken, there should be a correction for multiple testing on the level of the many outcome parameters, if there was not one a priori primary analysis? Or a MANOVA or something else (I'm not a statistician). Anyhow, the results are convincing, but the approach should be transparent.

Thank you for these two comments. Because the methods section comes at the end, these methodological details were not clarified already in the introduction. In brief, we analyzed data of each measure with a separate statistical analysis in line with our previous work with the same number of measures (Salehinejad et al., 2021, *Nature Communications*), which enables comparability between study results. Changing the approach would have made comparisons difficult. Moreover, the different parameters we tested were delivering information about different entities, i.e. these were informative in their own right, which in our opinion would make it tricky to put them all under one overarching hypothesis as precondition for correction for multiple comparisons overall conditions explored in the study. Finally, this strategy would end up in a situation where “small” studies testing only one or two parameters would much more likely result in significant results as compared to studies with multiple parameters, as done here, in case of identical numerical results, which would also not be helpful, and make comparisons between different study designs extremely problematic. For each dependent variable, mixed factorial or within-subjects ANOVAs were employed. And for each major analysis, correction for multiple comparisons was applied. We clarified this in detail in the methods section. Please see pages 32-34.

Methods pages 32-34:

“For the TMS protocols with a double-pulse condition (i.e., SICI-ICF, I-wave facilitation, SAI), the resulting mean values were normalized to the respective single-pulse condition. […] No correction for multiple comparisons was done for correlational analyses as these were secondary exploratory analyses.”

– I assume I will understand the design after reading the method section. However, on page 10 in the Results section, I still do not understand if there are parallel gropus or repeated measures or a combination of it.

For analysis of neuroplasticity data (on page 10) we had a parallel-group design and used a mixed factorial ANOVA with repeated measures (thus the combination of both) with sleep condition (normal vs deprivation) and timepoint as the within-subject factors and group (anodal vs cathodal) as between-subject factor. The respective analysis was thus a mixed-model ANOVA. This was clarified in the methods section and now also in the Results section. We also clarified this in the figure 1 legend in the introduction. Please see the response to the previous comment, second paragraph. We also added more details about the design in the introduction:

Revised text, Page 6:

“To do so, we recruited 30 healthy, right-handed participants in this randomized, cross-over study. All participants attended two experimental sessions after having sufficient sleep [23:00-8:00], or sleep deprivation [23:00-8:00]. All physiological, behavioral, and hormonal measures were obtained in each session (see Methods for details) at a fixed time. For the neuroplasticity measures, half of the participants received anodal and the other half cathodal stimulation in a randomized, sham-controlled parallel-group design. Figure 1 shows the detailed course of study. “

Revised text, Page 10:

“We analyzed the MEPs by a mixed-model ANOVA with stimulation condition (active, sham), time-point (7 levels), and sleep condition (normal vs deprivation) as within-subject factors and group (anodal vs cathodal) as between-subject factor. A significant four-way interaction of sleep condition×group×tDCS state×timepoint was found (*F*5.35=12.14, *p*<0.001, *η*p2=0.30), indicating that tDCS-induced LTP/LTD-like neuroplasticity was differentially affected in the sleep conditions. Other interactions and main effects are summarized in Table 1.”

– It appears to me that numerous correlation analyses do not represent the best way of further exploring the relationships between parameters. Other approached might be considered, such a linear regression models or others (I'm not a statistician, and linear regression might also be inappropriate).

Thank you for this comment. We need to clarify that our study was not sufficiently powered for a correlational design, and thus calculated correlations as a secondary aim of this work. We ran exploratory correlations between different measures to identify associations to be explored in detail with a larger sample size in future studies. We think that exploring associations with correlations in this exploratory approach makes sense content wise. Calculation of regressions would involve a couple of predictors, and thus be aimed to deliver statements about the relative contribution of single predictors, which might be even more problematic with the current sample. We added a respective statement to the results.

Revised text, page 10:

“Although our study was not sufficiently powered for conducting correlational analyses between measures as primary outcome, we ran exploratory correlation analyses to identify associations between physiologically and cognitive parameters which are conceptually related, including measures of plasticity and motor learning on the one hand, and parameters of cortical excitability and working. memory/attention on the other. Several relevant correlations between motor learning *and* plasticity, as well as working memory/attention *and* excitability indices were identified. LTP-like plasticity effects after sufficient sleep were correlated with better sequence learning acquisition (*r*=-0.558, *p*=0.031) and retention (*r*=-0.734, *p*=0.002). Enhanced working memory and sustained attention after sufficient sleep were also correlated with higher cortical facilitation and lower cortical inhibition (supplementary materials).”

Discussion:– I would expect that the beginning of the introduction more clearly relates to the tested model/hypotheses, that is provides the context, and not only lists findings (e.g., our findings provide further support for the synaptic homeostasis hypothesis…; or, in line with our initial hypothesis…); moreover, I would expect that the beginning of the introduction more clearly provides information about what represents a replication of prior work, an extension or a novel contribution to the literature.

Thank you for the suggestions. We conducted a major revision of the introducing paragraph of the discussion. Please see the changes below.

Revised text, page 19-20:

“In this study, we investigated how cortical excitability, brain stimulation-induced neuroplasticity and cognitive functions are affected by one-night sleep deprivation. […] Finally, changes in concentration of cortisol and melatonin were minor at best, in line with recent works (e.g. Kuhn et al., 2016) and cannot explain the observed effects.”

– There is some redundancy in the discussion, for instance the repetition of the introduction at the beginning of the third paragraph (Previous studies in animals…) that could be deleted.– On the other hand, it appears that the study provides first findings on LTD–like plasticity in humans – a novelty aspect that could be reemphasized in the third paragraph.– The finding of a conversion from LTD– to LTP–like plasticity after cathodal stimulation (after sleep deprivation) and the respective potential physiological explanation is of great interest.

Thank you for making these points. We rephrased the 3rd paragraph and removed redundant information. We also re-emphasized the LTD findings in a new paragraph. Please see the revised text below.

Revised text, page 18-19:

“Cortical excitability is a basic physiological response of cortical neurons to an input and is, therefore, a fundamental aspect of neuroplasticity, and cognition (Kuhn et al., 2016; Ly et al., 2016; Gaggioni et al., 2019). […] Critically, this conversion of LTD-like into LTP-like plasticity is in line with a saturating effect of sleep deprivation on synaptic strength rather than cortical under activation, which would be otherwise an alternative explanation for reduced LTP-like effects after anodal tDCS.”

– The authors should add that they cannot disentangle the contribution of different sleep stages or fine graded sleep processes, such as sleep slow waves or spindles.– The authors might add a paragraph of the potential relevance of their findings for healthy performance (e.g. high demanding jobs), aging, and disorders, such as major depression (e.g., synaptic plasticity model, Wolf et al. SMR) or cognitive decline in relation sleep/disrupted sleep.– Also, the idea of modulating sleep is currently of interest.

Thank you for making this point. We added a new paragraph in the discussion and discuss the mentioned points. Please see below:

Revised text p. 23-24:

“Our findings have several implications. First, they show that sleep and circadian preference (i.e., chronotype) have functionally different impacts on human brain physiology and cognition. […] Finally, sleep consists of different stages including REM, non-REM, slow-wave sleep, and spindles that can have a specific impact on plasticity, which could not be disentangled in our data.”

– I think the conclusion paragraph is too unspecific; rather specify, e.g. provide further support for… complement… provides first evidence for …

Thank you for making this point. We rephrased the whole concluding paragraph. Please see below:

Revised text p. 24:

“In conclusion, this experiment provides further evidence in humans supportive of the sleep homeostasis hypothesis. General information of upscaled brain excitability, as shown in previous works, was complemented by evidence that shows alterations in specific parameters of cortical excitability (e.g., increased facilitation, decreased and/or converted inhibition). The saturated state with respect to the inducibility of LTP-like plasticity aligns with the increased synaptic strength after sleep deprivation. The results of the present study also provide first evidence for LTD-like plasticity being converted into LTP under sleep pressure in the human brain, in conceptual accordance with an hyperexcitable state under sleep deprivation. These findings complement current knowledge about the critical role of sleep for neuroplasticity and cognition in humans.”

Methods:– Participants: Please provide information about potential substance use (including smoking).

We only included non-smokers and clarified this in the participants' section. Please see page 22. Also, in the methods section 4.3 we described that the use of CNS-active substances was not allowed (Page 25).

Revised text, page 25:

“All participants were right-handed non-smokers, with a regular sleep-wake pattern (determined by sleep diary) and underwent a medical screening to verify no history of neurological diseases, epilepsy or seizures, central nervous system-acting medication, metal implants, and current pregnancy.”

Revised text, Page 26:

“Food and drinks were provided (the consumption of coffee, caffeine-containing soft drinks, black tea, and alcohol was not allowed), watching TV programs, reading and working on the computer were also allowed. Participants were prevented from any sleep-related activities during the night, such as lying down, or closing their eyes for a prolonged time. They were refrained from drinking caffeine-containing drinks and sleeping or taking a nap from the afternoon before they joined the sleep deprivation session at 23:00.”

– Study design: It's the first time (on page 22) that I start to understand the design; please ensure including some fundamental information at an earlier section in the manuscript (such as end of the introduction, or beginning of the Results section).

Please see pages 4-5.

Revised text, p.4-6:

“In the sleep deprivation paradigm applied in the present study, participants are kept in an extended wakefulness condition for a certain amount of time. […] For the neuroplasticity measures, half of the participants received anodal and the other half cathodal stimulation in a randomized, sham-controlled parallel-group design. Figure 1 shows the detailed course of study.”

– Sleep conditions: did participants in the sleep condition slept in their home environment?

Yes, participants were instructed to sleep at home and follow our instructions for having a normal and sufficient sleep. In the case of non-compliance or self-reported sleep pressure in the sufficient sleep condition, the session was postponed. We clarified this in the methods section 4.3. Please see below.

Revised text, page 26:

“In the “sufficient sleep” condition, participants had to go to bed in their home environment at around 23:00 and have at least 8 hours of uninterrupted sleep. This was to prevent potential poor sleep in a new environment (i.e. the laboratory). The experiment was scheduled to start at 9:00 am. Participants were refrained from drinking alcohol and coffee 12 h before sleep time and afterwards until the end of the session. In the case of poor sleep quality for any reason (measured by sleepiness rating scales) or unregular sleep pattern (sleep onset, wake-up time) informed by the sleep diary (more than ±2.5 h deviation from the scheduled time frames), the respective session was canceled and postponed until sufficient sleep condition requirements were met.”

[Editors' note: further revisions were suggested prior to acceptance, as described below.]

The manuscript has been improved but there are some remaining issues that need to be addressed. While a comprehensive comparison and data integration across the different measures will likely not be possible with the current sample size, and the study can thus be viewed as a parallel investigation of different research questions within the same sample, please revisit the other core points of the previous editorial letter:Theta: the association between theta and synaptic strength remains unclear. The manuscript now makes this link less explicitly, however, would profit from a more dedicated attempt to clarify potential differences in the human vs. animal literature; the topographic origin of the observed theta findings (previous presentation suggested a stronger occipital rather than midfrontal effects as might be expected from the human literature); and the potential oscillatory nature of the findings (periodic vs. aperiodic).

Thank you for making this point. To properly address these core points, we added a new paragraph to the introduction (pages 4-5). In this revision we clarify that the theta oscillations were related most probably to sleep pressure rather than plasticity, and we distinguish between two types of theta, of which only sleep pressure related theta was our oscillatory activity of interest. For the topographic origin, our *focus* was rather on the midfrontal theta as it is closely associated with sleep pressure (Brown et al., 2012; Kuhn et al., 2016; Magnuson et al., 2022). Also, it is known that during sleep deprivation, α rhythms appear in posterior cortical recordings whilst theta rhythms increase in frontal cortical regions (Brown et al., 2012). The previous presentation (which is not included in the revision anymore, and thus in our opinion an extended discussion about the dominant topography might not be required) was showing the same pattern (i.e., stronger α and β rhythms in the posterior regions, while enhanced prefrontal theta). Please see pages 4-5. Revised text, pages 4-5 (introduction): In addition to these primary objectives, we were interested in brain oscillatory activities which are well-established indicators of the sleep-wake cycle and inform about the physiological state of the sleep-deprived brain. Specifically, theta oscillations are related to sleep and cognition. Here, at least two types of theta oscillations are distinguishable: one related to cognition and information processing, which occurs during wakefulness, but also REM sleep (Brown et al., 2012; Puentes-Mestril et al., 2019), and one related to sleep pressure due to extended wakefulness (Vyazovskiy and Tobler, 2005). For the former, animal studies show that it is generated by the hippocampus and involves mainly the temporal lobes at the level of the neocortex. In humans, where the temporal lobes are located ventrally, and thus difficult to record specifically from surface EEG, these theta rhythms with a strong regularity are observed mainly in frontal and midline cortices. The second type of theta, which is of main interest here, is of cortical origin, less regular, predominantly, but not exclusively observed over frontal-midline areas, and builds up with growing sleep pressure, including sleep deprivation, in both, animals and humans studies (Brown et al., 2012; Finelli et al., 2000; Magnuson et al., 2022; Snipes et al., 2022; Vyazovskiy and Tobler, 2005). While there is no clear and direct link between this sleep pressure-dependent theta activity and synaptic strength as suggested by some works, recent works in humans however linked it to an increase of cortical excitability (Kuhn et al., 2016). We addressed the functional implications of these findings also in the discussion. Please see page 23.

Revised text, page 23 (discussion).

“We also monitored EEG oscillatory activities and found an increase of theta-band activity in the frontal-midline regions during sleep deprivation. […] These analyses were thus not carried out here, however, they may provide new information about functional EEG markers during sleep deprivation related to plasticity and the capacity of information processing (Hanslmayr et al., 2016; Helfrich et al., 2021).”

Multiple comparisons correction: please state more explicitly how correction for multiple comparisons was performed, i.e. within or across analyses.

Thank you for the comment. We had conceptually related, but separate and independent datasets. Therefore, each data set was analyzed independently. Post hoc tests performed within each analysis were corrected for multiple comparisons. We clarify this in the methods section as follows: Revised text: 1.1. Statistical Analysis {Bushey, 2011 #3411}{Bushey, 2011 #3411}.

Data availability: data on the corresponding OSF site does not seem to include raw EEG data. In particular given the still lacking differentiation of periodic vs. aperiodic activity, making this data available would allow others to look into this issue themselves.

Thank you for the clarification. EEG raw data is now added to the OSF site and accessible via https://osf.io/ckph7/